# PET/CT based cross-modal deep learning signature to predict occult nodal metastasis in lung cancer

Yifan Zhong [1,10], Chuang Cai[2,10], Tao Chen[1,10], Hao Gui[3], Jiajun Deng[1], Minglei Yang[4], Bentong Yu[5], Yongxiang Song[6], Tingting Wang[7], Xiwen Sun[8], Jingyun Shi[8], Yangchun Chen[9], Dong Xie [1] ✉, Chang Chen [1] ✉ & Yunlang She[1] ✉

Occult nodal metastasis (ONM) plays a significant role in comprehensive treatments of non-small cell lung cancer (NSCLC). This study aims to develop a deep learning signature based on positron emission tomography/computed tomography to predict ONM of clinical stage N0 NSCLC. An internal cohort (n = 1911) is included to construct the deep learning nodal metastasis signature (DLNMS). Subsequently, an external cohort (n = 355) and a prospective cohort (n = 999) are utilized to fully validate the predictive performances of the DLNMS. Here, we show areas under the receiver operating characteristic curve of the DLNMS for occult N1 prediction are 0.958, 0.879 and 0.914 in the validation set, external cohort and prospective cohort, respectively, and for occult N2 prediction are 0.942, 0.875 and 0.919, respectively, which are significantly better than the single-modal deep learning models, clinical model and physicians. This study demonstrates that the DLNMS harbors the potential to predict ONM of clinical stage N0 NSCLC.

In the era of molecular imaging, positron emission tomography/computed tomography (PET/CT), which concurrently characterizes metabolic and anatomic representations about lesions, has emerged as the most dependable non-invasive modality for clinical N staging of non-small cell lung cancer (NSCLC)[1]. However, despite the tremendous advances in staging modality, there are still 12.9%–39.3%[2–4] of lymph nodal metastasis that are not identified by this state-of-the-art procedure and instead are unexpectedly recognized during surgery, which is defined as occult nodal metastasis (ONM).

Lymph node staging including N1 and N2 status plays a crucial role throughout the whole process of management for NSCLC. Hence, accurately recognizing ONM is critical in determining the optimal therapeutic strategies for patients with NSCLC. In a presurgical setting, nodal biopsy remains the gold-standard reference for defining the N stage of NSCLC. The routine adoption of this procedure, however, increases the risk of overdiagnosis, which is attributable to its invasive nature, and potentially leads to missed diagnosis considering the diagnostic pitfalls for N1 stations[5–7]. Accordingly, it is necessary to

[1]Department of Thoracic Surgery, Shanghai Pulmonary Hospital, Tongji University School of Medicine, Shanghai, China. [2]School of Computer Science and Communication Engineering, Jiangsu University, Zhenjiang, Jiangsu, China. [3]Graduate School at Shenzhen, Tsinghua University, Shenzhen, China. [4]Department of Thoracic Surgery, Ningbo HwaMei Hospital, Chinese Academy of Sciences, Zhejiang, China. [5]Department of Thoracic Surgery, The First Affiliated Hospital of Nanchang University, Jiangxi, China. [6]Department of Thoracic Surgery, Affiliated Hospital of Zunyi Medical University, Guizhou, China. [7]Department of Radiology, Zhongshan Hospital, Fudan University, Shanghai, China. [8]Department of Radiology, Shanghai Pulmonary Hospital, Tongji University School of Medicine, Shanghai, China. [9]Department of Nuclear Medicine, Shanghai Pulmonary Hospital, Tongji University School of Medicine, Shanghai, China. [10]These authors contributed equally: Yifan Zhong, Chuang Cai, Tao Chen. ✉e-mail: xiedong@tongji.edu.cn; changchenc@tongji.edu.cn; langthoracic@tongji.edu.cn

obtain the pretest probability of ONM to equipoise the superiority and inferiority of this dual-nature procedure.

In terms of surgical decisions, substantial evidence has emerged that sublobectomy and limited nodal dissection (LND), which preserves more of the lung parenchyma, could deliver comparable oncological efficacy to conventional lobectomy and systematic nodal dissection (SND) in early-stage NSCLC. However, tumors with nodal metastasis harbor a more aggressive behavior and greater malignancy burden, making that sublobectomy and LND insufficient. Therefore, lobectomy and SND should be conducted to ensure the adequacy of surgical margins and radicality of nodal removals[8–10].

In a postsurgical setting, the benefits of adjuvant therapy in early-stage NSCLC has been passionately debated[11,12]. The occurrence of nodal involvement heralds a more guarded prognosis[13] and thereby calls for more aggressive treatments[10]. For NSCLC with nodal metastasis, surgery alone cannot confer sufficient oncological efficacy, and adjuvant therapy, capable of eradicating the residual mircometastasis, has been demonstrated to provide additional survival benefits[14–19]. Therefore, it is of paramount importance to develop a robust instrument for ONM prediction to recognize candidates for nodal biopsy, lobectomy, SND and adjuvant therapy in clinical stage N0 NSCLC.

The deep learning technology which allows the high-dimensional quantification of radiological images and greater extraction of detailed characterizations than the human vision, has been proposed as a revolutionary approach for disease diagnoses, prognosis evaluations, and therapeutic decisions[20–22]. PET/CT, which is capable of capturing the anatomic and metabolic representations of tumors[23], has been leveraged as a dependable imaging modality to characterize malignancy grade and metastasis burden[24,25]. Its multimodal nature, on the one hand increases the feature dimensions and information abundance, but on the other hand poses a higher requirement for the deep learning algorithm.

With the development of multimodal algorithms, the current deep learning technology has evolved to be an effective method for PET/CT image analyzing[26,27], which harbors the capability of taking full advantages of the complementary information of PET and CT modalities. It has been demonstrated that multimodal deep learning algorithms shown potentials in cancer identification[28], tumor segmentation[29,30], and risk quantification[31] based on PET/CT imaging. Despite these tremendous breakthroughs, the application of PET/CT based deep learning for ONM prediction of lung cancer is limited. We hypothesize that cross-modal dominance complementation based on PET and CT imaging is capable of quantifying ONM probability to support the comprehensive treatments of clinical N0 NSCLC, and the captured ONM risks would be associated with histologic, genetic, and microenvironment behaviors.

Therefore, this study aims to combine PET and CT radiomics to construct a deep learning nodal metastasis signature (DLNMS) to predict ONM and personalize comprehensive treatments of clinical N0 NSCLC, and tentatively explore the underlying biologic basis of DLNMS, based on a large multicenter population.

## Results

### Study design and baseline information
The study design is described in Fig. 1. The baseline characteristics of the internal cohort, external cohort and prospective cohort are detailed in Table 1. The mean age of the entire cohort was 60.00 years and 48.61% ($n = 1587$) of the population were male. There were 2776 (85.02%) adenocarcinomas and 340 (10.41%) squamous cell carcinomas. The maximum standard uptake value (SUVmax), metabolic tumor volume (MTV), total lesion glycolysis (TLG) of the primary tumors were 5.43, 10.13 and 37.74, respectively. With respect to N status, 11.64% ($n = 380$) and 8.42% ($n = 275$) of patients were diagnosed as occult N1 and N2 diseases. In addition, compared to the internal cohort, patients in the external cohort were associated with significantly and older age

(61.78 years versus 59.42 years, $p < 0.001$) and patients in the prospective cohort yielded an older age (60.46 years versus 59.42 years, $p = 0.005$), higher SUVmax of primary tumor (5.67 versus 5.25, $p = 0.022$) and larger tumor size (2.64 cm versus 2.53 cm, $p = 0.030$).

### Variables associated with ONM
As displayed in Table 2, in the training set, a younger age (odds ratio [OR]: 0.967, 95% confidence interval [CI]: [0.951, 0.984], adjusted $p < 0.001$), pure solid type (OR: 2.525, 95% CI: [1.638, 3.891], adjusted $p < 0.001$), left location (OR: 1.512, 95% CI: [1.088, 2.100], adjusted $p = 0.023$), and central location (OR: 1.743, 95% CI: [1.202, 2.530], adjusted $p = 0.007$) were identified as independent predictors for occult N1 metastasis, and the pure solid type (OR: 3.389, 95% CI: [1.999, 5.745], adjusted $p < 0.001$) was independently related to occult N2 involvement. Most variables remained predictive for patients in the validation set, external cohort and prospective cohort (Supplementary Table 1). In addition, after incorporation of the DLNMS into analyses (Supplementary Table 2 & 3), the DLNMS was revealed as independent predictors for both occult N1 and N2 involvements.

### Predictive performance of DLNMS
With an increase of DLNMS scores, more cases with occult N1 and N2 tumors were observed in the validation set (Supplementary Fig. 1A &B), external cohort (Supplementary Fig. 1C & D) and prospective cohort (Supplementary Fig. 1E & F). In addition, the DLMNS was represented by conventional PET and CT texture features in ONM prediction, implying the significant correlations between the DLNMS and PET/CT texture features (Fig. 2).

As illustrated in Fig. 3A and B, Table 3 and Supplementary Fig. 2, in the validation set, the abilities of the DLNMS to predict occult N1 and N2 diseases were shown to have areas under the receive operating characteristic curve (AUROCs) of 0.958 (95% CI: [0.923, 0.992]) and 0.942 (95% CI: [0.911, 0.973]), respectively, which were significantly better than 0.873 (95% CI: [0.835, 0.911]) and 0.761 (95% CI: [0.680, 0.842]) of the PET model, 0.913 (95% CI: [0.875, 0.952]) and 0.887 (95% CI: [0.823, 0.952]) of the CT model, 0.752 (95% CI: [0.685, 0.819]) and 0.690 (95% CI: [0.603, 0.776]) of the clinical model, 0.612 (95% CI: [0.536, 0.689]) and 0.672 (95% CI: [0.574, 0.771]) of the senior physicians, and 0.616 (95% CI: [0.544, 0.687]) and 0.556 (95% CI: [0.465, 0.647]) of the junior physicians (DeLong's test: all $p < 0.05$). The areas under the precision-recall curve (AUPRC), sensitivity, specificity, positive predictive value (PPV), positive predictive value (NPV) and accuracy of the DLNMS for predicting occult N1 and N2 metastasis were 0.882, 0.898, 0.928, 0.647, 0.984 and 0.924, and 0.876, 0.897, 0.842, 0.317, 0.990, and 0.846, respectively.

In the external cohort (Fig. 3C, D), the DLNMS achieved AUROCs of 0.879 (95% CI: [0.813, 0.946]) and 0.875 (95% CI: [0.820, 0.930]) in predicting occult N1 and N2 metastasis, respectively, and were significantly superior than the PET model (0.790, 95% CI: [0.733, 0.847] and 0.727, 95% CI: [0.649, 0.805]), the CT model (0.826, 95% CI: [0.747, 0.905] and 0.817, 95% CI: [0.748, 0.887]), the clinical model (0.722, 95% CI: [0.642, 0.802] and 0.723, 95% CI: [0.648, 0.797]), the senior physicians (0.676, 95% CI: [0.590, 0.763] and 0.645, 95% CI: [0.554, 0.735]), and the junior physicians (0.633, 95% CI: [0.548, 0.719] and 0.594, 95% CI: [0.503, 0.685]) (DeLong's test: all $p < 0.05$). In addition, the AUPRC, sensitivity, specificity, PPV, NPV and accuracy of the DLNMS for predicting occult N1 and N2 metastasis were 0.853, 0.700, 0.905 0.483, 0.960 and 0.882, and 0.849, 0.857, 0.813, 0.333, 0.981, and 0.817, respectively.

In the prospective cohort (Fig. 3E, F), the DLNMS achieved AUROCs of 0.914 (95% CI: [0.877, 0.949]) and 0.919 (95% CI: [0.886, 0.942]) in discriminating occult N1 and N2 involvements, and were evidently better than the PET model (0.796, 95% CI: [0.751, 0.841] and 0.712, 95% CI: [0.656, 0.768]), the CT model (0.828, 95% CI: [0.777, 0.879] and 0.835, 95% CI: [0.779, 0.891]), the clinical model (0.749,

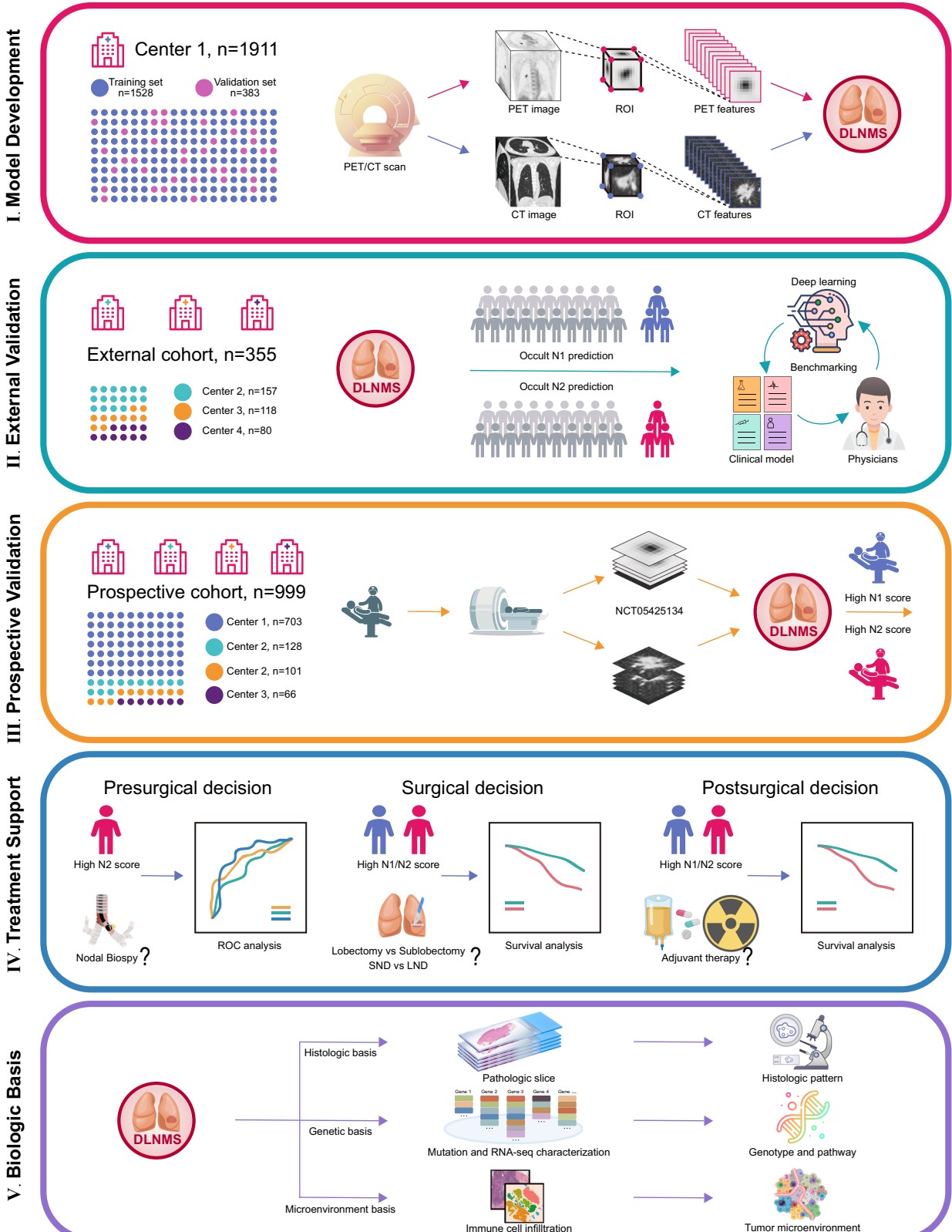

**Fig. 1 | Flow chart illustrating study design.** PET/CT, positron emission tomography-computed tomography; ROI, region of interest; DLNMS, deep learning nodal metastasis signature; SND, systematic nodal dissection; LND, limited nodal dissection; ROC, receiver operating characteristic curve.

95% CI: [0.708, 0.791] and 0.675, 95% CI: [0.629, 0.721]), the senior physicians (0.672, 95% CI: [0.623, 0.722] and 0.670, 95% CI: [0.613, 0.723]), and the junior physicians (0.645, 95% CI: [0.596, 0.693] and 0.635, 95% CI: [0.580, 0.691]) (DeLong's test: all $p < 0.05$).

Additionally, the AUPRC, sensitivity, specificity, PPV, NPV and accuracy of the DLNMS for occult N1 and N2 prediction were 0.871, 0.793, 0.926 0.586, 0.971 and 0.911, and 0.863, 0.833, 0.828, 0.308, 0.982, and 0.829, respectively.

**Table 1 | Baseline characteristics of patients in the internal cohort, external cohort and prospective cohort**

| Characteristics | Entire (n = 3265) | Internal cohort (n = 1911) | External cohort (n = 355) | Prospective cohort (n = 999) | p1 value | p2 value |
|---|---|---|---|---|---|---|
| Age (years) | | | | | | |
| >65, n (%) | 991 (30.35) | 524 (27.42) | 133 (37.46) | 334 (33.43) | <0.001 | 0.001 |
| ≤65, n (%) | 2274 (69.65) | 1387 (72.58) | 222 (62.54) | 665 (66.57) | | |
| Mean ± SD | 60.00 ± 9.31 | 59.42 ± 9.31 | 61.78 ± 8.71 | 60.46 ± 9.42 | <0.001 | 0.005 |
| Sex, n (%) | | | | | 0.949 | 0.985 |
| Male | 1587 (48.61) | 917 (47.99) | 171 (48.17) | 479 (47.95) | | |
| Female | 1678 (51.39) | 994 (52.01) | 184 (51.83) | 520 (52.05) | | |
| Smoking, n (%) | | | | | 0.986 | 0.858 |
| Ever | 491 (15.04) | 286 (14.97) | 53 (14.93) | 152 (15.22) | | |
| Never | 2774 (84.96) | 1625 (85.03) | 302 (85.07) | 847 (84.78) | | |
| Radiologic type, n (%) | | | | | 0.166 | 0.071 |
| Pure solid | 1860 (56.97) | 1060 (55.47) | 211 (59.44) | 589 (58.96) | | |
| Subsolid | 1405 (43.03) | 851 (44.53) | 144 (40.56) | 410 (41.04) | | |
| PET parameters | | | | | | |
| SUVmax mean ± SD | 5.43 ± 4.74 | 5.25 ± 4.69 | 5.72 ± 5.21 | 5.67 ± 4.66 | 0.086 | 0.022 |
| MTV, mean ± SD | 10.13 ± 15.62 | 9.84 ± 15.40 | 10.53 ± 12.36 | 10.53 ± 17.03 | 0.427 | 0.270 |
| TLG, mean ± SD | 37.74 ± 160.35 | 37.07 ± 153.29 | 37.73 ± 91.45 | 42.30 ± 190.10 | 0.938 | 0.422 |
| Surgery procedure, n (%) | | | | | 0.993 | 0.852 |
| Sublobectomy | 184 (5.64) | 105 (5.49) | 19 (5.35) | 60 (6.01) | | |
| Lobectomy | 3045 (93.26) | 1785 (93.41) | 332 (93.52) | 928 (92.89) | | |
| Pneumonectomy | 36 (1.10) | 21 (1.10) | 4 (1.13) | 11 (1.10) | | |
| Location, n (%) | | | | | | |
| Left | 1426 (43.68) | 803 (42.02) | 166 (46.76) | 457 (45.7) | 0.097 | 0.054 |
| Right | 1839 (56.32) | 1108 (57.98) | 189 (53.24) | 542 (54.3) | | |
| Central | 566 (17.34) | 346 (18.10) | 54 (15.20) | 166 (16.62) | 0.189 | 0.317 |
| Peripheral | 2699 (82.66) | 1565 (81.90) | 301 (84.80) | 833 (83.38) | | |
| Radiological size (cm), mean ± SD | 2.58 ± 1.26 | 2.53 ± 1.22 | 2.66 ± 1.39 | 2.64 ± 1.29 | 0.068 | 0.030 |
| N1 involvement, n (%) | | | | | 0.807 | 0.930 |
| Yes | 380 (11.64) | 224 (11.72) | 40 (11.27) | 116 (11.61) | | |
| No | 2885 (88.36) | 1687 (88.28) | 315 (88.73) | 883 (83.39) | | |
| N2 involvement, n (%) | | | | | 0.291 | 0.819 |
| Yes | 275 (8.42) | 156 (8.20) | 35 (9.90) | 84 (8.41) | | |
| No | 2990 (91.58) | 1755 (91.80) | 320 (90.10) | 915 (91.59) | | |
| Pathological type, n (%) | | | | | 0.764 | 0.566 |
| Adenocarcinoma | 2776 (85.02) | 1633 (85.45) | 302 (85.07) | 841 (84.18) | | |
| Squamous cell carcinoma | 340 (10.41) | 197 (10.31) | 35 (9.86) | 108 (10.81) | | |
| Others | 149 (4.56) | 81 (4.24) | 18 (5.07) | 50 (5.01) | | |

*PET*, positron emission tomography; *SUV*, standard uptake value; *MTV*, metabolic tumor volume; *TLG*, total lesion glycolysis; *SD*, standard deviation; p1 value for comparing the internal cohort with the external cohort; p2 value for comparing the internal cohort with the prospective cohort; categorical variables were analyzed by Pearson $\chi^2$ test and Fisher exact test, continuous variables were compared by Student t-test and Mann-Whitney U test.

In subgroup analyses regarding pathological types for patients in the validation set, external cohort and prospective cohort, the DLNMS achieved AUROCs of 0.916 (95% CI: [0.885, 0.947]) and 0.934 (95% CI: [0.915, 0.953]) in adenocarcinoma population for occult N1 and N2 prediction, respectively. Additionally, for squamous cell carcinoma population, the DLNMS yielded AUROCs of 0.904 (95% CI: [0.842, 0.966]) and 0.858 (95% CI: [0.779, 0.937]) for occult N1 and N2 prediction, respectively (Fig. 3G, H).

For patients in the validation set, external cohort and prospective cohort, the DLNMS could correct 38.30% occult N1, 73.11% benign N1, 78.13% occult N2, and 53.04% benign N2 diseases in those incorrectly diagnosed by the PET model (Supplementary Fig. 3A & B). Similarly, for those incorrectly predicted by the CT model, the DLNMS could correct 35.42% occult N1, 67.06% benign N1, 93.80% occult N2, and 41.18% benign N2 diseases (Supplementary Fig. 3C, D).

The calibration curves revealed that the DLNMS yielded good performances (Supplementary Fig. 4). Furthermore, we evaluated the clinical usefulness of the DLNMS compared to single-modal models for ONM detection via decision curve analyses, indicating that the DLNMS achieved better net benefits than other models no matter for occult N1 or N2 prediction (Supplementary Fig. 5). As summarized in Supplementary Table 4, the positive values of integrated discrimination improvements (all adjusted $p < 0.05$) and net reclassification index (all adjusted $p < 0.05$) for occult N1 and N2 predictions could be achieved when comparing the DLNMS to single-modal models.

**Decision support for nodal biopsy**

For 366 patients receiving nodal biopsy (Supplementary Table 5), the DLNMS yielded an AUROC of 0.853 (95% CI: [0.812, 0.895]) for predicting occult N2 diseases, which was significantly better than the PET

**Table 2 | Logistic analyses of occult N1 and N2 metastasis before incorporation of the DLNMS for patients in the training set**

| Variables | Occult N1 | | | | Occult N2 | | | |
|---|---|---|---|---|---|---|---|---|
| | Univariable | | Multivariable | | Univariable | | Multivariable | |
| | OR (95% CI) | p value | OR (95% CI) | adjusted p value | OR (95% CI) | p value | OR (95% CI) | adjusted p value |
| Age | 0.979 (0.963-0.994) | 0.008 | 0.967 (0.951-0.984) | <0.001 | 0.992 (0.973-1.012) | 0.445 | | |
| Sex (Male) | 1.929 (1.404-2.652) | <0.001 | 1.403 (1.001-1.990) | 0.066 | 1.463 (1.008-2.125) | 0.046 | 0.996 (0.670-1.482) | 0.985 |
| Smoking history (Ever) | 1.152 (0.824-1.470) | 0.855 | | | 1.301 (0.878-1.353) | 0.765 | | |
| Radiological type (Solid) | 4.005 (2.718-5.903) | <0.001 | 2.525 (1.638-3.891) | <0.001 | 4.231 (2.614-6.848) | <0.001 | 3.389 (1.999-5.745) | <0.001 |
| Location (Left) | 1.429 (1.048-1.949) | 0.024 | 1.512 (1.088-2.100) | 0.023 | 1.249 (0.862-1.810) | 0.241 | | |
| Location (Central) | 2.998 (2.146-4.188) | <0.001 | 1.743 (1.202-2.530) | 0.007 | 1.936 (1.282-2.924) | 0.002 | 1.240 (0.794-1.935) | 0.430 |
| Radiological size | 1.385 (1.239-1.548) | <0.001 | 1.146 (1.001-1.313) | 0.057 | 1.269 (1.124-1.433) | <0.001 | 1.144 (1.084-1.330) | 0.100 |
| SUVmax | 1.127 (1.095-1.160) | <0.001 | 1.045 (0.805-1.356) | 0.741 | 1.094 (1.060-1.129) | <0.001 | 0.862 (0.657-1.131) | 0.473 |
| MTV | 1.001 (0.993-1.010) | 0.746 | | | 0.999 (0.986-1.011) | 0.845 | | |
| TLG | 1.001 (1.000-1.002) | 0.105 | | | 1.000 (1.000-1.001) | 0.242 | | |

DLNMS, deep learning nodal metastasis signature; SUV, standard uptake value; MTV, metabolic tumor volume; TLG, total lesion glycolysis; HR, hazard ratio; CI, confidence interval; p values of multivariable analyses were corrected by the Benjamini and Hochberg method.

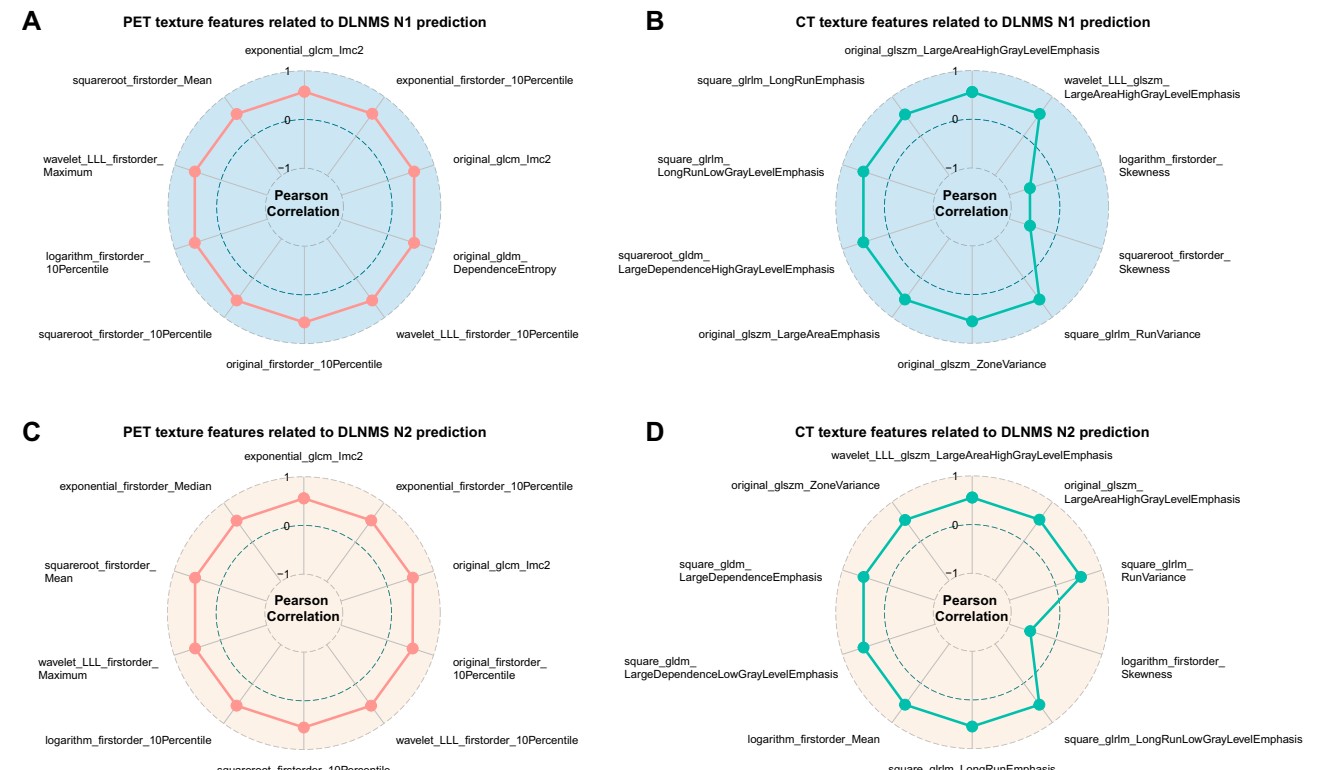

**Fig. 2 | PET and CT texture features related to the DLNMS. A** Top 10 PET and (**B**) top 10 CT texture features related to the DLNMS N1 prediction in the training set. **C** Top 10 PET and (**D**) top 10 CT texture features related to the DLNMS N2 prediction in the training set. $n = 1528$ biologically independent samples were examined. Source data are provided as a Source Data file. DLNMS, deep learning nodal metastasis signature; PET, positron emission tomography; CT, computed tomography.

model (0.644, 95% CI: [0.573, 0.715]), the CT model (0.780, 95% CI: [0.718, 0.841]), the clinical model (0.543, 95% CI: [0.471, 0.715]), the senior physicians (0.621, 95% CI: [0.554, 0.688]), and the junior physicians (0.525, 95% CI: [0.457, 0.594]). The AUPRC, sensitivity, specificity, PPV, NPV and accuracy of the DLNMS were 0.857, 0.919, 0.699, 0.436, 0.971 and 0.743, respectively (Fig. 4A & Table 3). In addition, with an increase in the DLNMS scores, more patients with occult N2 tumors were observed in the nodal biopsy cohort (Fig. 4B). Moreover, the DLNMS could correct 79.13% occult N2 and 56.41% benign N2 diseases in patients incorrectly diagnosed by the PET model (Fig. 4C). Similarly, for those incorrectly predicted by the CT model, the DLNMS could correct 100% occult N2 and 41.50% benign N2 diseases (Fig. 4D).

**Decision support for surgical treatment**
Survival analyses revealed that both N1 and N2 cutoff values could significantly stratify the prognosis of patients in the validation set and external cohort (Supplementary Fig. 6). In addition, patients with clinical stage I NSCLC (including patients receiving LND) were divided into low-risk (N1 score <0.362 and N2 score <0.356) and high-risk (N1 score > 0.362 or N2 score > 0.356) groups. The baseline characteristics of 654 clinical stage I patients receiving LND are provided in Supplementary Table 6. As illustrated in Fig. 5, for the low-risk population (Fig. 5A-D), sublobectomy did not compromise oncological results to lobectomy (3-year overall survival [OS]: 98.1% versus 97.4%, $p = 0.458$; 3-year recurrence-free survival [RFS]: 90.0% versus 90.6%, $p = 0.749$), and LND could achieve similar survival outcomes to SND (3-year OS: 98.1% versus 97.3%, $p = 0.428$; 3-year RFS: 90.4% versus 93.0%, $p = 0.965$). In contrast, for the high-risk population (Fig. 5E–H), patients receiving lobectomy yielded improved prognosis compared to those with sublobectomy (3-year OS: 90.9% versus 80.9%, $p = 0.011$;

3-year RFS: 79.0% versus 59.0%, $p < 0.001$) and SND conferred superior prognosis to LND (3-year OS: 91.7% versus 81.7%, $p = 0.008$; 3-year RFS: 79.2% versus 62.8%, $p = 0.001$).

**Decision support for adjuvant therapy**
As illustrated in Fig. 6, for patients diagnosed as pathological stage I NSCLC (including patients receiving LND), those without postoperative adjuvant therapy achieved comparable prognosis to those with postoperative adjuvant therapy in the low-risk group (3-year OS: 98.0% versus 97.5%, $p = 0.581$; 3-year RFS: 91.3% versus 89.3%, $p = 0.323$) (Fig. 6A & B). Conversely, in the high-risk group (Fig. 6C & D), patients receiving postoperative adjuvant therapy conferred significantly superior oncological results than those without postoperative adjuvant therapy (3-year OS: 95.9% versus 86.2%, $p = 0.034$; 3-year RFS: 90.5% versus 76.1%, $p = 0.012$).

**Biologic basis of DLNMS**
Both higher N1 and N2 scores were significantly related to the presence of aggressive histologic patterns including lymphovascular invasion (LVI), visceral pleural invasion (VPI), tumor spread through air space (STAS), micropapillary component, and solid component (all $p < 0.001$) (Fig. 7A, B). In addition, among patients with available data for common gene alternations, patients with high N1 scores were significantly relevant to the higher frequency of BRAF mutation ($p < 0.001$) and larger proportion of AKL mutation ($p = 0.004$) (Fig. 7C). Patients with high N2 scores yielded a significantly lower mutation rate of EGFR ($p < 0.001$) (Fig. 7D). In the gene set enrichment analysis (GSEA) and single sample gene set enrichment analysis (ssGSEA) analysis (Fig. 7E–G), pathways related to tumors proliferation such as signaling by GPCR, NTRKs and WNT in cancer were

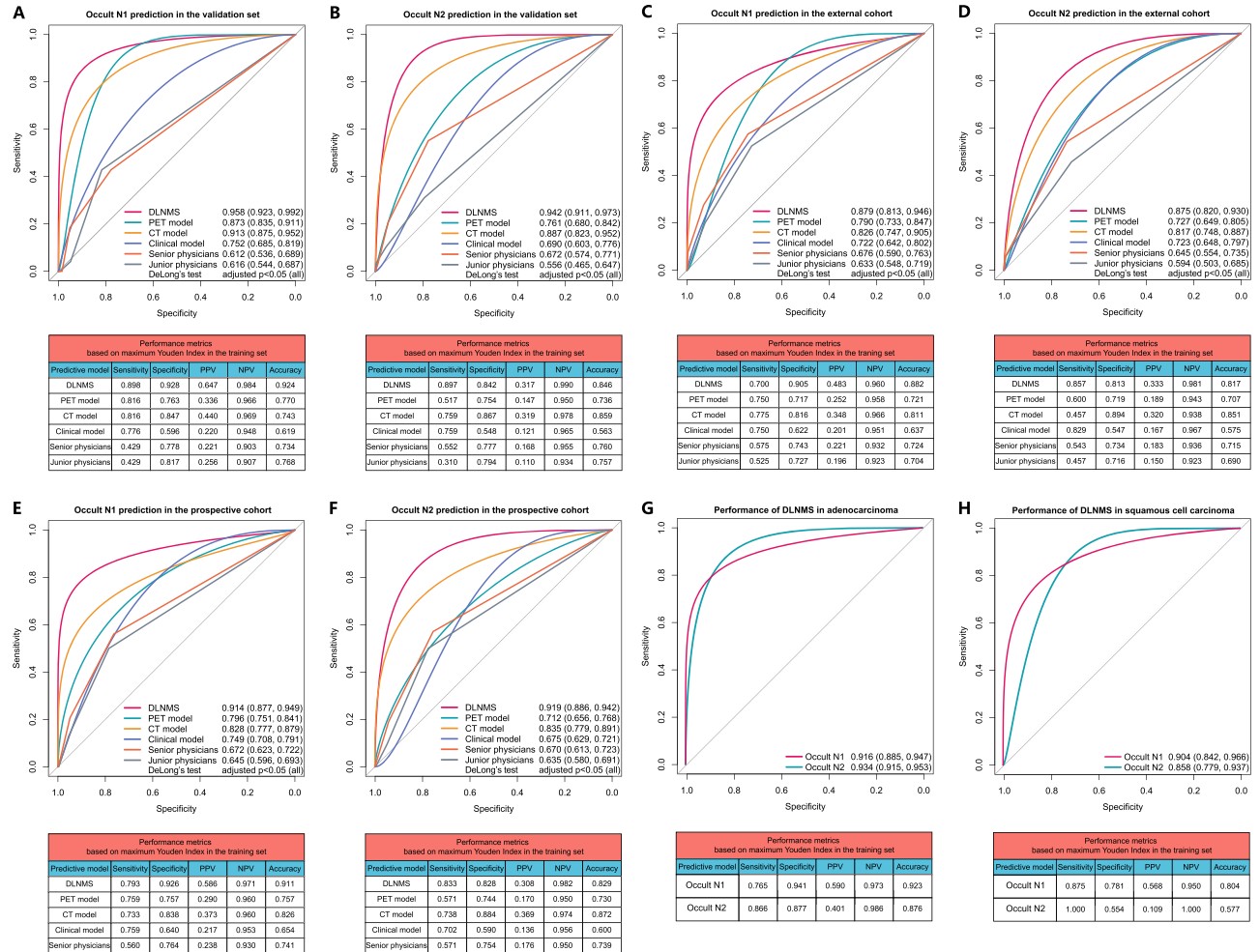

**Fig. 3 | Predictive performances of the DLNMS for occult nodal metastasis in clinical stage N0 non-small cell lung cancer.** ROC curves and performance metrics of models to predict occult N1 and N2 in the (**A**, **B**) validation set, **C**, **D** External cohort and (**E**, **F**) prospective cohort. ROC curves and performance metrics of the DLNMS to predict occult nodal metastasis in (**G**) adenocarcinoma and (**H**) squamous cell carcinoma for patients in validation set, external cohort and prospective cohort. n = 383, 355, and 999 biologically independent samples were examined for the validation set, external cohort, and prospective cohort, respectively. p values from Delong's tests were adjusted by the Benjamini and Hochberg corrections for 5 multiple comparisons. Source data are provided as a Source Data file. ROC, Receiver operating characteristic curve; DLNMS, deep learning nodal metastasis signature; PPV, positive predictive value; NPV, negative predictive value; PET, positron emission tomography; CT, computed tomography.

**Table 3 | Areas under precision-recall curves of different models for occult N1 and N2 prediction**

| Models | Occult N1 prediction | | | Occult N2 prediction | | | |
|---|---|---|---|---|---|---|---|
| | Validation set | External cohort | Prospective cohort | Validation set | External cohort | Prospective cohort | Biopsy cohort |
| DLNMS | 0.882 | 0.853 | 0.871 | 0.876 | 0.849 | 0.863 | 0.857 |
| PET model | 0.756 | 0.731 | 0.748 | 0.753 | 0.710 | 0.741 | 0.752 |
| CT model | 0.779 | 0.751 | 0.764 | 0.765 | 0.746 | 0.755 | 0.761 |
| Clinical model | 0.656 | 0.612 | 0.627 | 0.694 | 0.635 | 0.648 | 0.695 |
| Senior physicians | 0.504 | 0.562 | 0.538 | 0.563 | 0.583 | 0.569 | 0.550 |
| Junior physicians | 0.582 | 0.590 | 0.599 | 0.514 | 0.555 | 0.534 | 0.514 |

*DLNMS* deep learning nodal metastasis signature; *PET* positron emission tomography; *CT* computed tomography.

significantly unregulated in patients with high N1 and N2 scores. Finally, in the analyses of the tumor microenvironments, tumors with high N1 scores showed more infiltrations of central memory CD4 T cells, mast cells and plasmacytoid dendritic cells. High N2 scores were significantly associated with greater proportions of central memory CD4 T cells and central memory CD8 T cells (Fig. 7H).

## Discussion

Preoperative nodal staging is a critical determinant for individualized treatments of patients with NSCLC[10]. For clinical stage N0 NSCLC, the occurrence of ONM would reduce the theoretical benefits of the initial treatments, therefore inadvertently excluding patients from optimal therapeutic strategies. In this regard, obtaining an accurate pretest

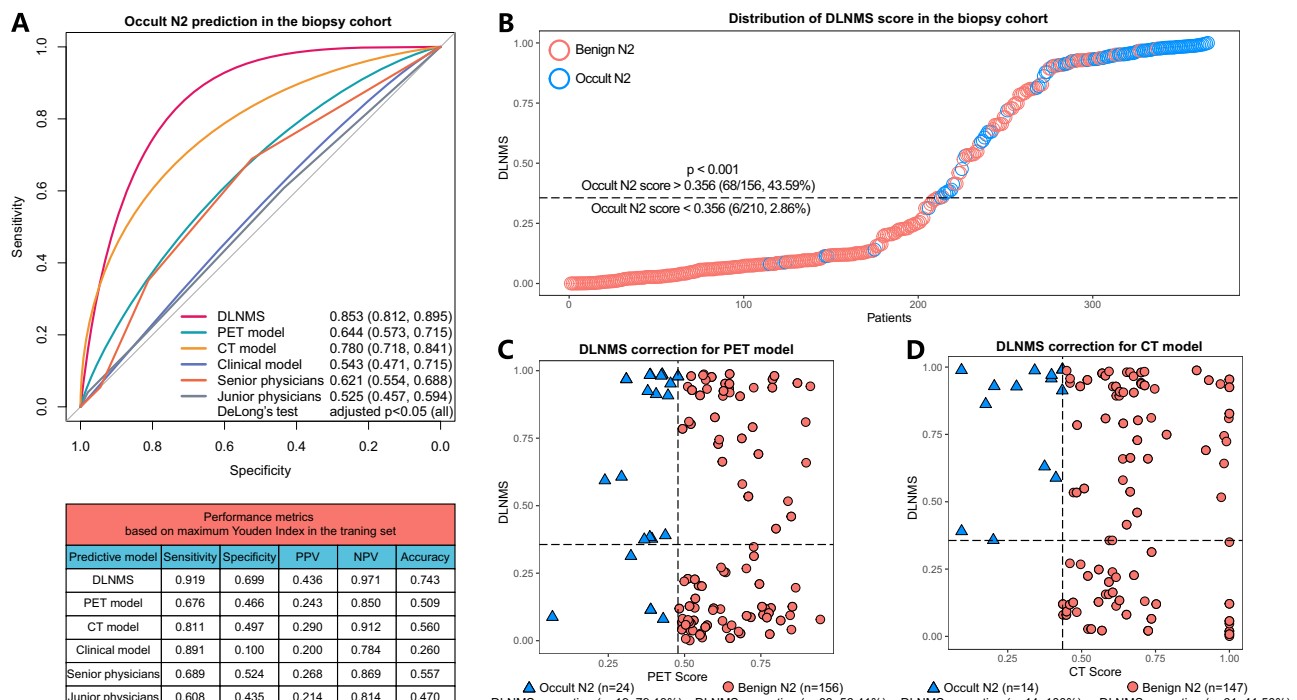

**Fig. 4 | Predictive performances of the DLNMS for occult N2 metastasis diagnosed by nodal biopsy in clinical stage N0 non-small cell lung cancer. A** ROC curves and performance metrics of models to predict occult N2 diagnosed by nodal biopsy. **B** Scatter graphs illustrating the DLNMS score distributions. **C, D** Scatter graphs describing the DLNMS correct cases falsely predicted by the PET and CT models. *n* = 366 biologically independent samples were examined. p values from Delong's tests were adjusted by the Benjamini and Hochberg corrections for 5 multiple comparisons. Source data are provided as a Source Data file. ROC, Receiver operating characteristic curve; DLNMS, deep learning nodal metastasis signature; PPV, positive predictive value; NPV, negative predictive value; PET, positron emission tomography; CT, computed tomography.

probability of ONM prior to treatments is of paramount importance. The current study managed to develop a cross-modal deep learning signature based on PET/CT images. The proposed DLNMS achieved AUROCs of 0.958, 0.879 and 0.914 for occult N1 prediction, and 0.942, 0.875 and 0.919 for occult N2 prediction, in the validation set, external cohort and prospective cohort, respectively. Moreover, high-risk patients defined by the DLNMS could benefit from nodal biopsy, lobectomy, SND and adjuvant therapy.

In clinical practice, clinical physicians mainly rely on certain clinical characteristics especially imaging features to capture the ONM risks of clinical stage N0 NSCLC. Evidences have emerged that metabolic and morphologic parameters on PET/CT, such as tumor size, central location, consolidation ratio, and metabolic value might provide efficient clues for ONM diseases[32–35]. Nevertheless, this subjective evaluation yields low AUROCs of 0.525-0.676 due to heterogenous experiences among physicians, and is incapable of comprehensively estimating the probability of ONM, so as to convey a direct implication to the management strategy for a given patient. The triumph of individually quantifying ONM risks based on predictive models represented a crucial step. Predictive rules integrating clinical variables could calculate the probability of ONM involvement in clinical N0 NSCLC. However, in spite of their higher accuracies than clinical physicians, these clinical models were far from meeting clinical requirements, resulting in AUROCs of 0.700-0.756[36–38], which was also observed by the current study, our clinical model only yielded AUROCs of 0.675-0.794 for ONM identification. As such, more valuable radiographic features for predicting ONM should be investigated to achieve clinical utility.

Radiomics, which allows quantitative extraction of high-dimensional radiological features, has provided a promising approach for more accurate evaluation of the lymph node status of lung cancer. Several studies have been successful in recognizing ONM

in early-stage NSCLC utilizing radiomics phenotypes, which yielded AUROC values of 0.808 to 0.820[39–41]. Despite such inspiring success, the above radiomics studies were limited in the CT modal, and the added value of PET radiomics features for ONM prediction of NSCLC are still ambiguous. With the development of multimodal algorithms, the deep learning approach has been applied to analyze PET/CT imaging[26–31]. Based on the main advancements of deep learning technology, multimodal fusion primarily involved three strategies: input-level concatenation[42,43], feature-level combination[44], and output-level average[45]. Our preliminary experiments investigated multiple deep learning architectures and fusion strategies, revealing feature-level fusion based on the ResNet 18[46] backbone yield better efficiencies and was finally utilized to generate our DLNMS. The current study demonstrated that the cross-modal DLNMS incorporating PET and CT radiomics features achieved AUROCs of 0.875-0.958, make it superior to single-modal models based on PET or CT alone for ONM prediction.

In the domain of machine learning, one issue worth mentioning is the method for performance evaluation. On an imbalanced dataset with a low proportion of positive classifications, the PR curve might be more effective than the ROC curve in quantifying positive discriminative ability[47,48]. However, what needs to be emphasized is that the PR curve only focuses on the efficiency to identify diseased cases but ignores those correctly predicted healthy cases[49]. Different from conventional classification tasks, ONM recognition would pose a direct impact on treatment decisions, which emphasizes model's discriminative abilities for both positive and negative subjects. If a patient diagnosed as healthy actually is ONM disease (false negative), this patient would directly lose the opportunity of receiving optimal treatments. The AUPRC is a summary indicator comprehensively quantifying the positive and negative predictive capabilities[50], we therefore chose the Youden Index based on ROC curves to determine the cutoff values of DLNMS.

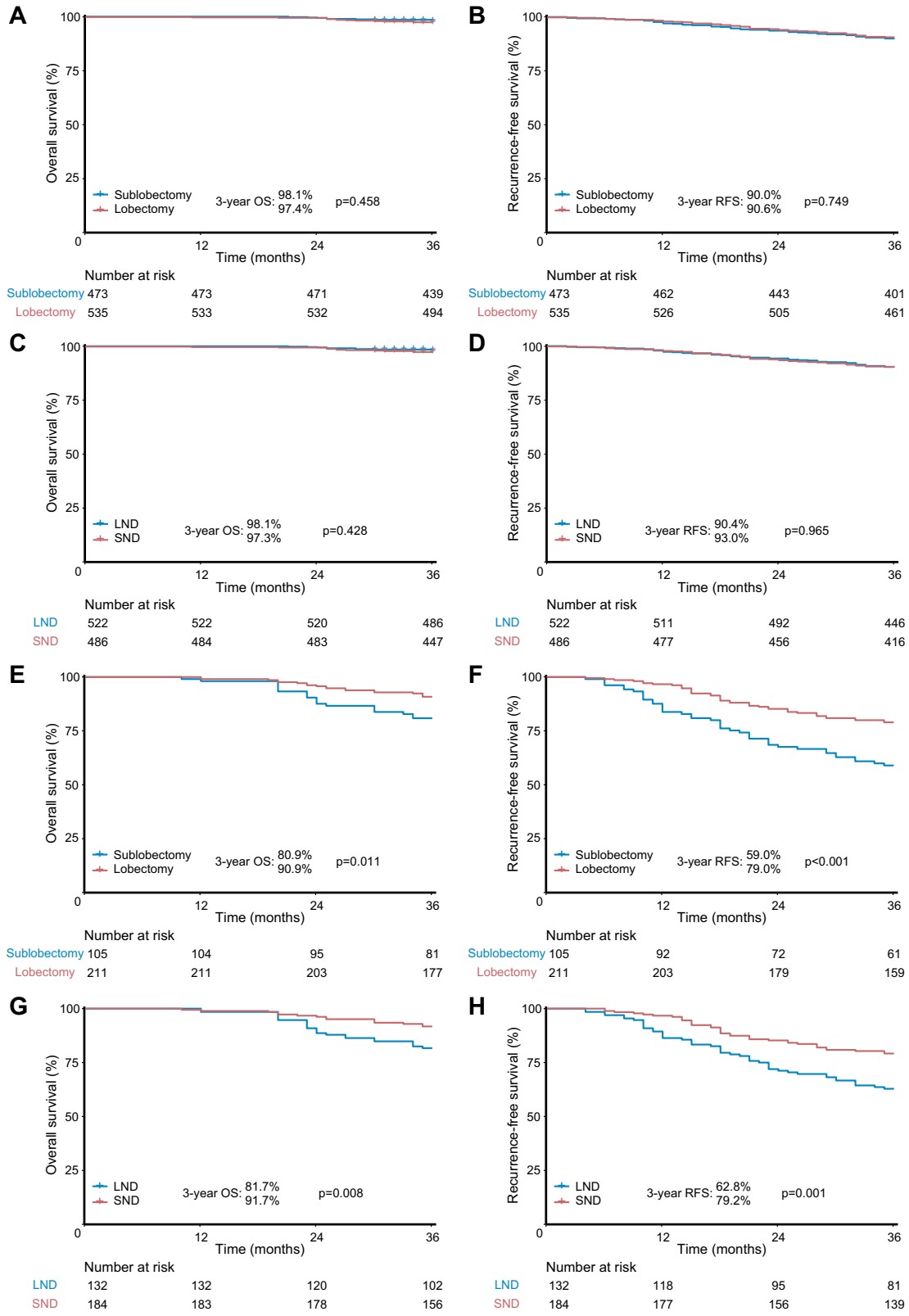

**Fig. 5 | Prognosis of clinical stage I non-small cell lung cancer treated with different surgical strategies for low-risk and high-risk patients in the validation set and external cohort.** Survival comparisons between (**A**, **B**) sublobectomy versus lobectomy and (**C**, **D**) LND versus SND in low-risk patients. Survival comparisons between (**E**, **F**) sublobectomy versus lobectomy and (**G**, **H**) LND versus SND in high-risk patients. $n = 1324$ biologically independent samples were examined. Survival data were compared by the log-rank test. Source data are provided as a Source Data file. SND, systematic nodal dissection; LND, limited nodal dissection; OS, overall survival; RFS, recurrence-free survival.

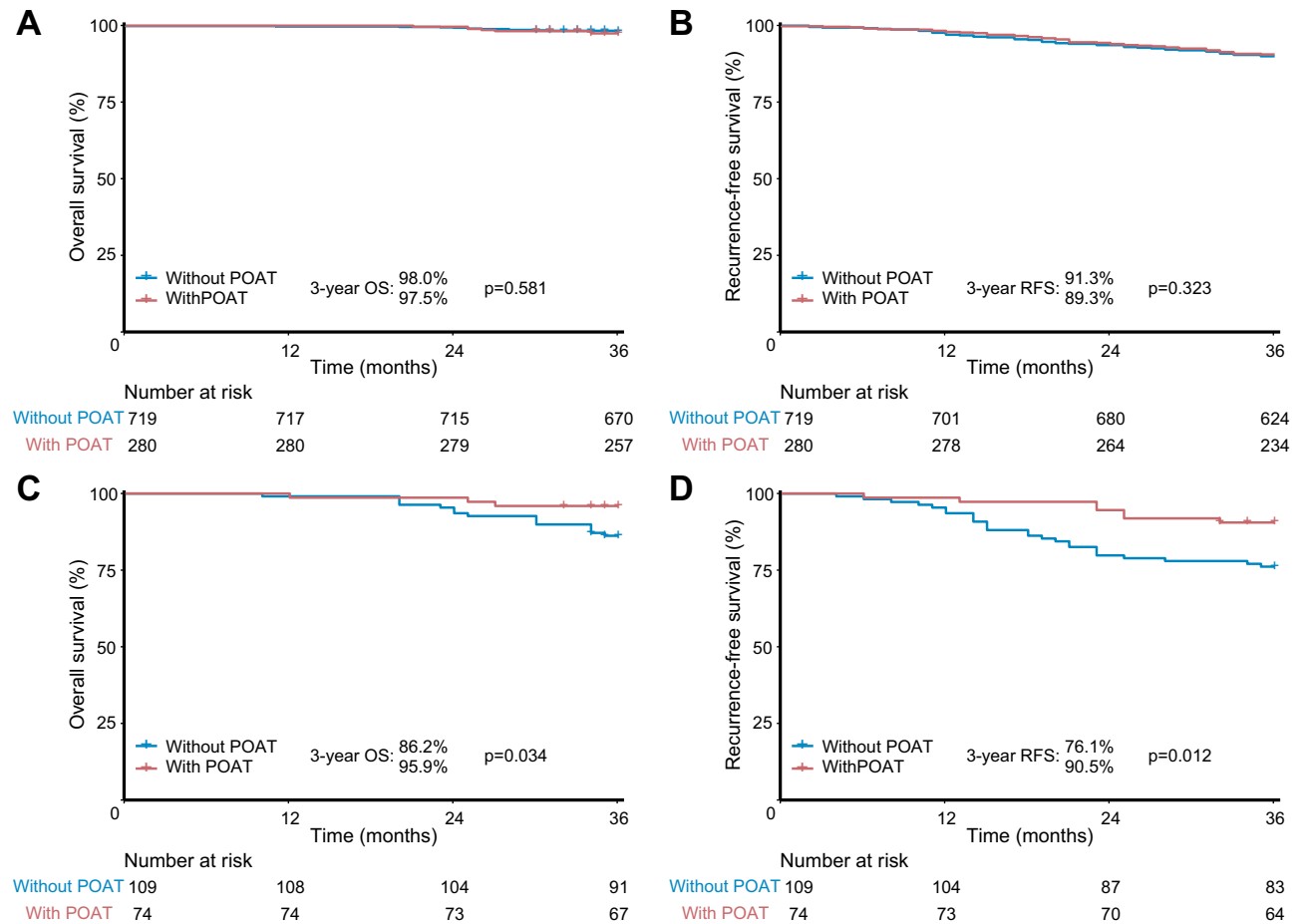

**Fig. 6 | Prognosis of pathological stage I non-small cell lung cancer with adjuvant therapy and without adjuvant therapy for low-risk and high-risk patients in the validation set and external cohort.** Survival comparisons between with adjuvant therapy versus without adjuvant therapy in (**A**) and (**B**) low-risk and (**C**) and (**D**) high-risk patients. $n = 1182$ biologically independent samples were examined. Survival data were compared by the log-rank test. Source data are provided as a Source Data file. POAT, postoperative adjuvant therapy; OS, overall survival; RFS, recurrence-free survival.

Whether a radiomics signature can be introduced into the clinical workflow to optimize the treatment decision is the benchmark for demonstrating its clinical utility. Distinguished from other radiomics studies that are limited in model constructions and efficiency evaluations, the current study took a further step to elucidate the potential application scenarios of the proposed DLNMS. In a presurgical setting, nodal biopsy serves as the gold-standard reference for N2 staging, but concurrently suffers from its invasive nature, thus emphasizing the necessity of equipoising the superiority and inferiority of this dual-nature procedure to individualize the N2 staging of NSCLC[5–7]. The DLNMS maintained efficiencies in the nodal biopsy population, therefore sparing patients with low N2 scores from this invasive procedure and ensuring that patients with high N2 scores receive nodal biopsy for adequate N2 staging. Additionally, for surgical decisions, sublobectomy and LND, with more lung preserves than conventional lobectomy and SND, have been increasingly adopted in the surgical treatment of clinical stage I NSCLC. However, if ONM occurs, lobectomy and SND are more appropriate choices[8–10]. Our results demonstrated that sublobectomy and LND were effective for patients with low ONM risks, while lobectomy and SND were mandatory in patients with high ONM risks to achieve the oncological radicality. Finally, in a postsurgical setting, adjuvant therapy eradicates the residual micrometastatic disease, but simultaneously has significant side effects, thus calling for appropriate patient selection to identify candidates for this double-edged sword[11,12]. Based on our results, patients with low risks

would not benefit from adjuvant therapy. In contrast, adjuvant therapy conferred survival superiority in patients with high risks.

Several limitations of this study should be acknowledged. Firstly, as a retrospective study, selection bias was inevitable, despite the inclusion of a prospective cohort for validation, and whether our findings are applicable to other territories remains unknown. To be confirmed, an international clinical trial is required. Secondly, the main histology of included cases were adenocarcinomas, and different histologies are represented by discrepant radiological phenotypes and tumor aggressiveness, contributing to their heterogeneity in the metastasis nature. Thus, a future study with adequate sample sizes in histologic subgroups should be conducted to validate the efficiency of the DLNMS. Thirdly, high-resolution CT findings is necessary to analyze the subtle images, however, not all PET/CT equipment harbor the capability of outputting such high-quality images, which might reduce the clinical applicability of the DLNMS in certain institutions. Finally, the main limitation of the current deep learning technique regarding medical imaging analyses is that, its black-box setting has the problem of interpretability. Despite our exploration of the biologic basis of the DLNMS, its working rationale was ambiguous and the predictive features were nameless. Therefore, studies deciphering the opaqueness of deep learning features in future is warranted.

In conclusion, the developed DLNMS is reliable in predicting ONM of clinical stage N0 NSCLC. Furthermore, the DLNMS has potentials for guiding individualized decisions for nodal biopsy, surgery and adjuvant therapy in clinical stage N0 NSCLC.

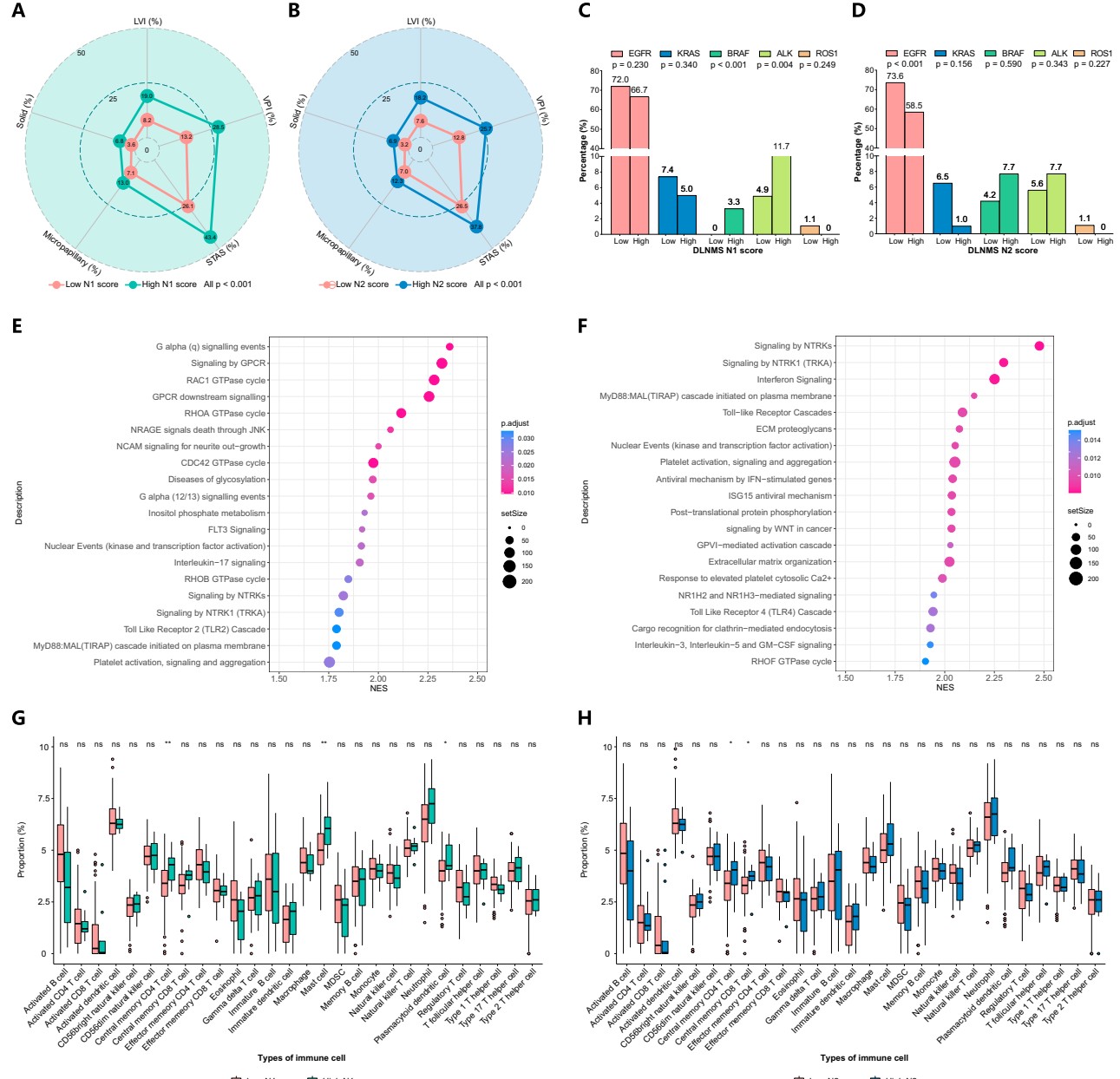

**Fig. 7 | Biologic basis of the DLNMS. A, B** Radar charts illustrating histologic patterns between low-score and high-score patients. **C, D** Bar charts showing frequency of gene alternations between patients with low scores and high scores. **E, F** Dot plots showing the top 20 upregulated molecular pathways in patients with high scores, p values were adjusted by the Benjamini and Hochberg corrections. **G, H** Boxplots comparing proportions of infiltrated immune cells between low-score and high-score patients. The centre of box denotes the 50th percentile, the bounds of box contain the 25th to 75th percentiles, the whiskers mark the maximum and minimum values, values beyond these upper and lower whiskers are considered outliers and marked with dots. n = 144 biologically independent samples were examined. Source data are provided as a Source Data file. DLNMS, deep learning nodal metastasis signature; LVI, lymphovascular invasion; VPI, visceral pleural invasion; STAS, tumor spread through air space; NES, normal enrichment score; EGFR, epidermal growth factor receptor; KRAS, kirsten ratsarcoma viral oncogene homolog; BRAF, v-raf murine sarcoma viral oncogene homolog B1; ALK, anaplastic lymphoma kinase; ROS1, c-ros oncogene 1; MDSC, myeloid-derived suppressor cells.

## Methods
### Study design and participants
This study was implemented under the approval of the Institutional Review Board of Shanghai Pulmonary Hospital, The First Affiliated Hospital of Nanchang University, Affiliated Hospital of Zunyi Medical College and Ningbo HwaMei Hospital. Written informed consent was waived for the internal and external cohorts and acquired for the prospective cohort. The study design is described in Fig. 1. The DLNMS was developed using an internal cohort (entire: n = 1911, occult N1 proportion = 11.64%, occult N2 proportion = 8.42%; training: n = 1528,

occult N1 proportion = 11.45%, occult N2 proportion = 8.31%; validation: n = 383, occult N1 proportion = 12.79%, occult N2 proportion = 7.57%). Subsequently, a multicenter external cohort (n = 355, occult N1 proportion = 11.27%, occult N2 proportion = 9.90%) and a multicenter prospective cohort (n = 999, occult N1 proportion = 11.64%, occult N2 proportion = 8.41%; ClinicalTrials.gov, NCT05425134) were adopted to fully validate the predictive efficiencies of the DLNMS by benchmarking the single-modal deep learning model, clinical model and physicians. Moreover, the values of the DLNMS for guiding nodal biopsy, surgery and adjuvant therapy decision-makings were explored via

efficiency evaluations in a nodal biopsy cohort ($n = 366$) and survival stratifications on different risk groups. Finally, the biologic basis of DLNMS was investigated by comparing histologic patterns, common genetic alternations, genetic pathways, and infiltrations of immune cells in microenvironments between patients with low and high scores. Patient selection details are provided in Supplementary Method 1 and Supplementary Fig. 7.

### Data acquisition and deep learning algorithm
Clinical information was retrieved from medical records, and follow-up data were acquired from outpatient visits and telephone interviews. The pathologic nodal status in the internal cohort, external cohort and prospective cohort was defined based on surgically resected specimens and that in the nodal biopsy cohort was defined based on nodal biopsy specimens. SND was defined as dissected N2 stations ≥ 3 with complete N1 dissection according to National Comprehensive Cancer Network guidelines[10]. Follow-up protocol details are described in Supplementary Method 2. The region of interest of the primary tumor was annotated by a junior thoracic radiologist (T.W., with 5 years of experiences) and confirmed by an expert thoracic radiologist (J.S., with 25 years of experiences). Details regarding the parameters of PET/CT scanners and tumor annotation are summarized in Supplementary Method 3 & 4. The structure of the DLNMS was illustrated in Supplementary Fig. 8. Two ResNet18 backbone networks[46] with the same structure were used to extract features from PET and CT images separately. Then, the PET and CT features were fused using the concat operation and input into a fully connected layer for classifications of ONM. The DLNMS consisted of two separate models predicting occult N1 and N2, respectively. For occult N1 prediction, data were divided into N1 metastasis and non-N1 metastasis. Similarly, in N2 prediction, data were divided into N2 metastasis and non-N2 metastasis. Details of image preprocessing and model construction procedures are provided in Supplementary Method 5-8. All computer codes for preprocessing and training are summarized at https://github.com/zhongthoracic/DLNMS.

### Cutoff calculation
Based on the maximum Youden index in the training set, the cutoff values of all models were determined to calculate the performance metrics and define the risk groups. The cutoff values of the DLNMS for occult N1 and N2 were calculated as 0.362 and 0.356, respectively. Therefore, patients with N1 scores > 0.362 and <0.362 were considered to have high and low occult N1 probabilities, respectively, and those with N2 scores > 0.356 and <0.356 were considered to have high and low occult N2 probabilities, respectively. Finally, by combining the N1 and N2 scores, patients were divided into high-risk (N1 scores > 0.362 or N2 scores > 0.356) and low-risk (N1 scores < 0.362 and N2 scores < 0.356) groups.

### Benchmarking
The predictive efficiency of the DLNMS was compared to the PET model, CT model, clinical model, senior physicians and junior physicians. The PET model and CT model were developed by the deep leaning algorithm based on the PET modality and CT modality, respectively. The clinical model was constructed by logistic analyses on the training set. For physicians, 3 senior radiologists and 3 junior radiologists blinded to pathological information were required to classify the ONM status based on imaging data. Benchmarking details are summarized in Supplementary Method 9.

### Comprehensive treatments support
For nodal biopsy decisions, the predictive efficiency and performance metrics of the DLNMS in the nodal biopsy cohort were evaluated. For surgery decisions of clinical stage I NSCLC, ONM risks for patients receiving LND were predicted by the generated DLNMS and included

into analyses (Supplementary Method 10). The prognosis of patients receiving lobectomy versus sublobectomy and SND versus LND was compared between the DLNMS defined low-risk and high-risk groups, respectively. For adjuvant therapy decisions of pathological stage I NSCLC, the oncological results of patients receiving adjuvant therapy versus not receiving adjuvant therapy were compared between the low-risk and high-risk groups.

### Biologic basis exploration
According to the cutoff values, distributions of patients with N1 scores < 0.362 versus N1 scores > 0.362 and N2 scores < 0.356 versus N2 scores > 0.356 in aggressive histologic patterns (LVI, STAS, VPI, micropapillary component, and solid component) and common genetic alternations (EGFR, KRAS, BRAF, ALK, and ROS1) were compared, respectively. Additionally, based on patients with NSCLC in the radiogenomics dataset (a public dataset comprising paired PET/CT and RNA sequencing data, https://wiki.cancerimagingarchive.net), the GSEA and ssGSEA were implemented to reveal heterogeneity in genetic pathways and infiltration of immune cells in tumor microenvironment between patients with different ONM scores. GSEA and ssGSEA procedures are detailed in Supplementary Method 11 & 12.

### Statistical analysis
Categorical variables were analyzed by Pearson χ2 test and Fisher exact test, continuous variables were compared by Student t-test and Mann-Whitney U test. The clinical model was generated based on the logistic regression analyses using a p-value level of 0·1. Survival data were assessed using the Kaplan-Meier method, log-rank test and Cox regression analyses. Predictive efficiency was evaluated by the AUROC and AUPRC. AUROCs among models were compared using the Delong's test. Performance metrics containing sensitivity, specificity, accuracy, PPV, and NPV were generated based on cutoff values determined by the maximum Youden index in the training set. CIs were computed by 10, 000 bootstrap replicates. The Benjamini and Hochberg method was utilized to correct p values from multiple comparisons. Analyses mentioned above was conducted using SPSS (version 25.0, IBM SPSS Statistics) and R program (version 4.1.3, http://www.Rproject.org). A $p < 0·05$ was regarded as having statistical significance.

### Reporting summary
Further information on research design is available in the Nature Portfolio Reporting Summary linked to this article.

## Data availability
The PET/CT imaging data in the current study are not publicly available for patient privacy purposes. However, if researchers wish to access our data solely for scientific research purposes and are willing to sign a data transfer agreement, the corresponding author can share the relevant data. Source data are provided with this paper.

## Code availability
Are provided at GitHub (https://github.com/zhongthoracic/DLNMS).

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

## Acknowledgements

This study was supported by National Natural Science Foundation of China (92259205, 82102126, 82272943); National Key Research and Development Program of China (2022YFC2407401); Science and Technology Commission of Shanghai Municipality (21YF1438200); Clinical Research Foundation of Shanghai Pulmonary Hospital (SKPY2021008); Investigator-Initiated Trial of Shanghai Pulmonary Hospital (2021LY1144, 2023LY0310); Ningbo Top Medical and Health Research Program (2022030208); and Medicine and Public Health Scientific Projects in Zhejiang Province (2020KY270). In addition, we would like to thank all members in the MultiomIcs claSSIfier for pulmOnary Nodules (MISSION) Collaborative Group for their supports and efforts.

## Author contributions

Y.Z., C.C., T.C., D.X., Y.S., and C.C. designed this study and wrote the paper. C.C. and H.G. built the deep learning models. J.D., T.W., X.S., J.S., and Y.C. processed and analyzed the data. M.Y., B.Y., and Y.S. collected the clinical dataset and performed data preprocessing. D.X., Y.S., and C.C. conceived the project and edited the paper. All authors reviewed and approved the final manuscript for submission. We ensured that all authors had access to all the raw datasets. Y.Z., C.C., and T.C. have verified the data and are independent of any company or investor. D.X., Y.S., and C.C. had full access to all the data in the study and had final responsibility for the decision to submit for publication.

## Competing interests

The authors declare no competing interests.
