## [Peer Review File · Nature Communications]

PET/CT based cross-modal deep learning signature to predict occult nodal metastasis in lung cancerREVIEWER COMMENTS

Reviewer #1 (Remarks to the Author): clinical expertise in lung cancer radiomics

In this study, the method of Radiomics analysis was used to evaluate the biological aggressiveness of lung cancer. PET/CT findings of the primary lesions including SUVmax, MTV, and TLG were extensively analyzed by deep learning as well as texture features obtained from CT findings. There are several important issues to be more discussed shown below.

(1) The values of SUVmax and other parameters by PET should vary among PET scanner models. What kind of adjustments were made among multiple facilities (different PET system) to standardize the measurement conditions? This is related to the application of DLNMS to cases in other facilities.

(2) The test set includes more than 10% squamous cell carcinoma cases. The SUVmax of adenocarcinoma is basically different from that of squamous cell carcinoma even though the stage is same. The authors did not pay attention to this matter therefore the results may be inaccurate.

(3) We believe that texture features obtained from CT findings are also used in the analysis. I would like to encourage authors to explain features analyzed.

(4) High-resolution CT findings is necessary to analyze the subtle images therefore it is difficult to obtain such precise findings using the conventional CT images associated with PET examination. Comments should be added.

Reviewer #2 (Remarks to the Author): expertise in neural networks in digital imaging

PET/CT based cross-modal deep learning signature to predict occult nodal metastasis and support comprehensive treatments in clinical stage N0 non-small cell lung cancer: a multicenter study

NCOMMS-23-11103

The study describes a validation of a deep learning model trained on PET/CT radiomics data for prediction of N1/N2 occult nodal metastasis in N0-staged patients. Training and testing are performed on multicenter multi-cohort data, with additional analyses of the model's value for clinical decision making and exploration of the biological processes associated with DLNMS predictions.

1. What are the noteworthy results?

The main strength of the study is substantial assessment of DLNMS prediction performance on multicenter multi-cohort data that required a lot of effort and delivered a very promising result. Specifically, it's impressive that the proposed DLNMS demonstrated the ability to outperform existing clinical models, radiologist evaluations, and single modality models on internal, external, prospective, and biopsy validation sets. The authors described cohorts in detail and ran additional univariate and multivariate feature analyses. These results indicate the model's robustness, generalizability, and potential for clinical utility. On top of that, it's great to see an extra effort towards assessing the utility of the model for (pre/post) surgical decision-making and exploring the "biological basis" of DLNMS predictions.

Overall, the main text is written well, albeit quite dense, especially because many details are tucked away in Supplemental Materials and aren't easy to find (e.g. positive/negative ratios for train/val sets are only available in Figure S1). Despite format limitations, I still encourage the authors to make an effort in improving the text flow and make more specific references to supplemental materials.

2. Will the work be of significance to the field and related fields? How does it compare to the established literature? If the work is not original, please provide relevant references.

The work demonstrates a promising approach to predicting occult nodal metastasis in early-stage

NSCLC patients. The use of multimodal deep learning in this context could potentially improve treatment decisions, leading to better patient outcomes. The DLNMS model's superior performance compared to clinical models and radiologist evaluations is a significant contribution to the literature on nodal metastasis prediction in NSCLC patients.

However, the study could benefit from a more thorough comparison with existing literature and methods. The authors review established literature very briefly and selectively and do not describe the state-of-the-art, which leads to a poorly defined knowledge gap that this study is trying to bridge. Although there may be no prior applications of deep learning to specifically predict NSCLC ONM, there is rich literature on related topics, from using conventional image processing techniques to high-performing multimodal deep learning for related tasks on PET/CT images (PMID: 36029345, PMID: 32773965, PMID: 33591922). E.g., see (PMID: 36870800) for overview of over a dozen studies that applied AI to PET/CT imaging data for management of lymphoma. Even if there is no previous method that DLNMS can be compared to, context for the general use of AI for similar tasks should be provided to establish the novelty and significance of the study in the context of the broader field.

3. Does the work support the conclusions and claims, or is additional evidence needed?

In general, the work supports the conclusions and claims made, as the authors provided robust evidence from multiple cohorts and validation sets. However, there are some issues with data analysis and methodology that need to be addressed before this article can be published – please see corresponding sections below.

4. Are there any flaws in the data analysis, interpretation and conclusions? Do these prohibit publication or require revision?

Overall, analyses, interpretation, and conclusions appear to be sound. The authors also point out some of the study limitations in Discussion.

One issue that still needs to be addressed is that the study seems to lack proper multiple testing correction for the reported significance results. There are many tests performed that are used to compare prediction approaches, variables, etc. increasing the risk of false discoveries. The authors should make sure to apply appropriate correction methods to account for this issue and ensure the validity of their findings. The correction procedures should be clearly described in the Statistical analysis subsection.

I have to point out that throughout the manuscript, the authors keep referring to various values of AUC as “satisfactory” or “unsatisfactory” predictive performances. Such assessment should be either substantiated by quantitative definition of what is “satisfactory” and why or should be avoided altogether.

Finally, while the authors did compare the performance of the DLNMS model with traditional clinical decision-making tools and human expert performance, it would be helpful to expand on these comparisons in the discussion section. This could involve exploring potential reasons for the observed differences in performance and discussing the implications of these findings for clinical practice.

5. Is the methodology sound? Does the work meet the expected standards in your field?

The methodology seems generally sound, however, I have some concerns with regard to the choice of hyperparameters and the rationale behind their selection. Providing a clear explanation of these choices is crucial for ensuring that the study adheres to the high standards expected in the field.

- The task setting is not clear. Data labeling and model training descriptions indicate that the model performs binary classification, however, separate scores are reported for N1 and N2 ONM (Figure 2A,B). Does that mean that 2 separate models were trained to predict N0 vs N1 and N0 vs N2? The authors should clarify data splits and model training very explicitly.

- Another issue that is not discussed enough is substantial class imbalance (1/9 if N1/N2 are considered separately), which the authors only briefly mention in data preprocessing. While the authors balance classes during training, it does affect prediction performance as demonstrated by PPV values (Figures 2, 3) and confusion matrices (Figure S3). When such class imbalance is present, AUC curves can be misleading and AUPR curves should be preferred (PMID: 25738806). I suggest the authors to either report AUPR curves next to AUC curves, or replace the latter with the former and move AUC curves to supplemental materials. Correspondingly, Results and/or Discussion should mention this issue, as in some cases the model predicts as many (Figure S3 E) or even +30% more false positives compared to true positives (Figure S3 3G).

- The main contribution of this study is assessment of the proposed model. However, any graphical or textual description of the model is missing from the main text. The authors should at least add a brief description of key characteristics to the main text and a diagram as a supplemental figure.

- The choice of many hyperparameters for constructing and training the model is not justified. For example, choosing ResNet-18 as the base architecture for the DLNMS model may be suboptimal, considering that it is relatively shallow and outdated compared to more recent deep learning architectures. Did authors consider alternative models with superior performance in various image recognition tasks, e.g. deeper ResNet, EfficientNet, or DenseNet?

- There are multiple important details missing from the description of the model in Supplemental Materials: (1) It's not clear how the multimodal PET/CT model was constructed, i.e. were two networks trained together with the final classifier, or was it just a combination of features from the two single-modality extractor (which is not truly multimodal)? Did the authors consider other options such as multi-task learning? (2) It should be explicitly stated that ResNet-18 the authors employed was pretrained on the ImageNet dataset of natural images. (3) There is no information on how image augmentations were selected, did the authors perform any ablation studies to make sure selected transforms do not hurt the performance? (4) There is no analysis of model computational requirements or convergence. (5) Supplementary Materials mention that the authors used binary cross-entropy as a loss function, while the code repository employs Focal loss instead.

Addressing the issues above will provide valuable insights into the model's inner workings and help to identify potential areas for further improvement.

6. Is there enough detail provided in the methods for the work to be reproduced?

Methods section need to be updated according to critiques above in order to allow other researchers to reproduce the method. Nevertheless, the authors could further enhance reproducibility by providing pre-trained models as supplementary material.

Minor issues:

- square brackets are missing around some confidence intervals
- please add subtitles in Figure 2, panels G-J; Figure 3, panels A-D
- please proofread Supplementary Information, there are many typos (e.g., line 74: retrospective - > retrospectively)
- Supplementary Information should have a separate References section and cite methods and packages used to perform the analyses (e.g. ResNet-18, ImageNet, etc.)

RESPONSE TO REVIEWERS' COMMENTS

We would like to express our gratitude to each of the external reviewers for careful and thorough reading of this manuscript and for the thoughtful comments and constructive suggestions, which help to improve the quality of this study. The comments are encouraging and the reviewers appear to share our judgement that this study and its results are clinically important. Please see below, in **blue**, our detailed response to reviewers' comments (comments are in *italics*). All mentioned line numbers refer to the manuscript file with tracked changes. We hope the revised manuscript is acceptable for publication in *Nature Communications*

Reviewer 1:

1. The values of SUVmax and other parameters by PET should vary among PET scanner models. What kind of adjustments were made among multiple facilities (different PET system) to standardize the measurement conditions? This is related to the application of DLNMS to cases in other facilities.

Reply:

We thank the reviewer for this important comment. We agree with you that adjustments for multiple facilities are critical for the application of DLNMS to cases in other facilities. In this study, we utilized three main methods to standardize the input images. Firstly, for the original PET/CT images, the energy attenuation and imaging noise are heterogeneous among multiple

facilities, which would affect the accuracy of quantitative analysis. Therefore, the attenuation correction was utilized to reduce statistical noises of original PET/CT images and the corrected images were reconstructed for quantitative analyses, which would preliminary reduce bias and improve accuracy of the diagnostic process. Secondly, to correct variability from image-acquisition techniques and reconstruction parameters related to voxel size, each voxel in the original images was resampled to $0.6 \times 0.6 \times 0.6$ mm³ in the spatial dimension. Through the resample process, images from different facilities were uniformed in the voxel size, which would reduce the variability from multicentre collection and different protocols. Finally, to reduce the batch effect error, pixel-wise PET/CT image data were normalized by z-score normalization, which means the ROI image is subtracted by the mean intensity value and divided by the standard deviation of the ROI image intensity. Via above three data pre-processing procedures, radiomics data harmonization among multiple facilities was improved to support following modelling and validation procedures. In this revision, data pre-processing procedures were detailed in Supplementary Method 8.

Changes in the text:

Supplementary Method 8. Adjustments among multiple facilities

Three methods were utilized to standardize images. Firstly, for the original PET/CT images, the attenuation correction was applied to reduce statistical noises of original images and the corrected images were reconstructed for quantitative analyses. Secondly, to correct variability

related to voxel size, each voxel in images was resampled to $0.6 \times 0.6 \times 0.6$ mm³ in the spatial dimension. Finally, to reduce the batch effect error, pixel-wise ROI image data were subtracted by the mean intensity value and divided by the standard deviation of ROI image intensity.

2. The test set includes more than 10% squamous cell carcinoma cases. The SUVmax of adenocarcinoma is basically different from that of squamous cell carcinoma even though the stage is same. The authors did not pay attention to this matter therefore the results may be inaccurate.

Reply:

Thank you for this significant question. We agree with you that the pathological type is a key characteristic of tumors. Squamous cell carcinomas may have different ONM risk compared to adenocarcinomas. In the current study, the main reason to include squamous cell carcinomas was to meet the clinical workflow. We can hypothesize a prospective real clinical scenario, in which a patient with pulmonary nodule is seeking the medical attention. Doctors could administrate the PET/CT scanning to this patient and define benign or malignancy of the detected lesion. The ability of PET/CT in identifying malignancy is supported by that PET/CT yielded an accuracy of about 90% in diagnosing pulmonary nodules [1-3] and PET/CT has been widely applied by doctors for diagnosing pulmonary nodules in clinical practice. Thereafter, if this patient is diagnosed as benign, follow-up visit could be suggested and unnecessary surgery could be avoided. Conversely, if this patient is diagnosed as

malignancy, DLNMS could be applied to these patients and further guide nodal biopsy, surgery and adjuvant therapy decisions. In summary, before surgery, we can determine if a patient with a pulmonary nodule is benign or malignant via PET/CT, but we cannot determine if the malignant nodule is adenocarcinoma or squamous cell carcinoma because that the precise pathological type of malignant nodule relies on primary lesion biopsy and this invasive biopsy has a low success rate for early lung cancer. In addition, in clinical practice, biopsy for primary lesion is also rarely conducted for early lung cancer. Therefore, if our DLNMS was constructed only for adenocarcinoma, it cannot be applied in the real clinical scenario considering we cannot determine whether a malignant nodule is adenocarcinoma or squamous cell carcinoma before surgery. In such instances, we must include squamous cell carcinoma into the targets of the DLNMS to meet the clinical workflow, which is mandatory for applying DLNMS to clinical practice in the future.

Furthermore, your concern on the accuracy of the DLNMS for squamous cell carcinoma is quite important and we agree with you that validating the DLNMS in different pathologies could further improve the robustness of the DLNMS and the quality of this paper. To investigate this, we retrieved all 195 squamous cell carcinomas in the internal validation set, external cohort and prospective cohort, the DLNMS achieved AUCs of 0.904 and 0.858 for N1 and N2 prediction in patients with squamous cell carcinomas (NEW Figure 3H). To further clarify this in an independent squamous cell carcinoma dataset, we further collected 322 squamous cell carcinomas from July 2022 to June 2023 in Shanghai Pulmonary Hospital, The First Affiliated Hospital of Nanchang University, Affiliated Hospital of Zunyi Medical College and Ningbo

HwaMei Hospital. Participants were included when meeting the following criteria: (a) patients receiving curative surgery for primary squamous cell carcinomas; (b) the maximum short-axis diameter of N1 and N2 nodes less than 1 cm on CT scan; (c) the SUVmax of N1 and N2 lymph nodes less than 2.5. Criteria for exclusion included (a) multiple lung lesions; (b) poor quality of PET/CT images; (c) patient not receiving SND; (d) patient receiving neoadjuvant therapy. The baseline information was detailed in Table-response 1. The N1 and N2 metastasis rate of the whole cohort was 22.0 and 10.9%. As illustrated in Figure-response 1, the proposed DLNMS maintained a satisfactory predictive efficacy, achieving AUCs of 0.910 and 0.856 for N1 and N2 prediction, respectively, which indicated that the proposed DLNMS could also be used in a larger population with squamous cell carcinomas.

Finally, we definitely agree with you that the SUVmax of adenocarcinoma is basically different from that of squamous cell carcinoma even though the stage is same. The PET and CT features of adenocarcinoma are both different from that of squamous cell carcinoma. Since that, the discrepancy in imaging features could also be learnt by the DLNMS. For example, squamous cell carcinoma was represented by higher SUVmax, higher CT density and larger size, when the DLNMS recognized tumors with these features, it tends to give a more N1 and N2 score for this population. To confirm this, we analyzed the deep learning scores between adenocarcinomas and squamous cell carcinomas, indicating that squamous cell carcinomas did have higher deep learning score than adenocarcinoma (entire cohort) (N1 score: 0.39 ± 0.34 versus 0.15 ± 0.25 , $p < 0.001$; N2 score: 0.42 ± 0.39 versus 0.18 ± 0.33 , $p < 0.001$). Therefore, the DLNMS did learnt nodal metastasis risks whether for adenocarcinomas or

squamous cell carcinomas, and the differences of imaging features between adenocarcinomas or squamous cell carcinomas are reflected by different risk scores between adenocarcinomas or squamous cell carcinomas. We think this point is the theoretical basis of the DLNMS predicting nodal metastasis risks in squamous cell carcinomas.

In this revision, we have provided the efficiency of the DLNMS in adenocarcinomas and squamous cell carcinomas subgroups based on existing data (New Figure 3G&H). However, the manuscript already contained too much content and the journal required strict word limits. Therefore, the independent squamous cell carcinomas dataset between July 2022 and June 2023 was only utilized to answer this question in the response letter and we did not add this cohort as a part of the research (Table-response 1 and Figure-response 1). Despite this, what needs to be emphasized is that the sample size of squamous cell carcinomas was relatively small and the population cohort was mainly based on Chinese population. Therefore, the predictive value of the proposed signature should be further validated in international multicenter population. Moreover, the sample size of other pathological types except for adenocarcinoma and squamous cell carcinomas were too small, so we describe this as a limitation of this study, and we hope the efficiency of our signature in different pathological subgroups could be further validated by larger study in future.

References

[1] Cronin P, Dwamena BA, Kelly AM, Carlos RC. Solitary pulmonary nodules: meta-analytic comparison of cross-sectional imaging modalities for diagnosis of malignancy. *Radiology*. 2008 Mar;246(3):772-82.

[2] Ohno Y, Koyama H, Matsumoto K, Onishi Y, Takenaka D, Fujisawa Y, Yoshikawa T, Konishi M, Maniwa Y, Nishimura Y, Ito T, Sugimura K. Differentiation of malignant and benign pulmonary nodules with quantitative first-pass 320-detector row perfusion CT versus FDG PET/CT. *Radiology*. 2011 Feb;258(2):599-609.

[3] Basso Dias A, Zanon M, Altmayer S, Sartori Pacini G, Henz Concatto N, Watte G, Garcez A, Mohammed TL, Verma N, Medeiros T, Marchiori E, Irion K, Hochegger B. Fluorine 18-FDG PET/CT and Diffusion-weighted MRI for Malignant versus Benign Pulmonary Lesions: A Meta-Analysis. *Radiology*. 2019 Feb;290(2):525-534.

Table-response 1. Baseline characteristics of patients with squamous cell carcinomas from

July 2022 to June 2023

Characteristics	Entire cohort (n=322)
Age (years)	
>65, n (%)	136 (42.2)
≤65, n (%)	139 (57.8)
Mean ± SD	63.9 ± 7.7
Sex, n (%)	
Male	311 (96.6)
Female	11 (3.4)
Smoking, n (%)	
Ever	113 (35.1)
Never	209 (64.9)
Location, n (%)	
Left	145 (45.0)
Right	177 (55.0)
Central	171 (53.1)
Peripheral	151 (46.9)
Tumor size (cm), mean ± SD	3.28 ± 1.46
N1 involvement, n (%)	71 (22.0)
N2 involvement, n (%)	35 (10.9)

Figure-response 1. Predictive performance of the DLNMS in patients with squamous cell carcinomas

Changes in the text:

Result section Line 192-197: In subgroup analyses regarding pathological types for patients in the validation set, external cohort and prospective cohort, the DLNMS achieved AUROCs of 0.916 (95% CI: [0.885, 0.947]) and 0.934 (95% CI: [0.915, 0.953]) in adenocarcinoma population for occult N1 and N2 prediction, respectively. Additionally, for squamous cell carcinoma population, the DLNMS yielded AUROCs of 0.904 (95% CI: [0.842, 0.966]) and 0.858 (95% CI: [0.779, 0.937]) for occult N1 and N2 prediction, respectively (Fig. 3G & H).

Discussion section Line 403-407: Secondly, the main histology of included cases were adenocarcinomas, and different histologies are represented by discrepant radiological phenotypes and tumor aggressiveness, contributing to their heterogeneity in the metastasis nature. Thus, a future study with adequate sample sizes in histologic subgroups should be conducted to validate the efficiency of the DLNMS.

New Fig. 3G&H. Predictive performance of the DLNMS in ADC and SCC in the current study

3. We believe that texture features obtained from CT findings are also used in the analysis. I would like to encourage authors to explain features analyzed.

Reply:

We appreciate the reviewer for these significant questions. In this study, PET and CT features were both extracted by the deep learning algorithm. Compared to conventional texture features based radiomics, the advantage of deep learning is that it can mine more deeper features than conventional radiomics with no need for manual segmentations and the constructed deep learning model is usually more accurate. Therefore, deep learning is more effective than conventional radiomics especially for large datasets. Meanwhile, the

disadvantage of deep learning is that extracted deep learning features are nameless and we cannot know the definition of each feature. Therefore, the low interpretability is the main drawback of deep learning. Your suggestion does inspire us a lot, we realize that we can explain our model from texture features to further improve the interpretability of the DLNMS. However, the texture features of PET/CT were not directly used and analysed in model construction in this study, we therefore calculated the correlations between the DLNMS and each PET/CT texture feature to identify top 10 key PET/CT texture features related to our DLNMS.

In this revision, correlation charts for PET features and correlation charts for CT features (Figure 2) were added to explain the DLNMS from texture features, indicating that for occult N1 prediction, the DLNMS was related to exponential_glcm_Imc2, exponential_firstorder_10Percentile, original_glcm_Imc2, original_gldm_DependenceEntropy, wavelet_LLL_firstorder_10Percentile, original_firstorder_10Percentile, squareroot_firstorder_10Percentile, logarithm_firstorder_10Percentile, wavelet_LLL_firstorder_Maximum and squareroot_firstorder_Mean in PET modality (Figure 2A) and original_glszm_LargeAreaHighGrayLevelEmphasis, wavelet_LLL_glszm_LargeAreaHighGrayLevelEmphasis, logarithm_firstorder_Skewness, squareroot_firstorder_Skewness, square_glrIm_RunVariance, original_glszm_ZoneVariance, original_glszm_LargeAreaEmphasis, squareroot_gldm_LargeDependenceHighGrayLevelEmphasis, square_glrIm_LongRunLowGrayLevelEmphasis and square_glrIm_LongRunEmphasis in CT

modality (Figure 2B). Similarly, for occult N2 prediction, the DLNMS was related to exponential_glcm_Imc2, exponential_firstorder_10Percentile, original_glcm_Imc2, original_firstorder_10Percentile, wavelet_LLL_firstorder_10Percentile, squareroot_firstorder_10Percentile, logarithm_firstorder_10Percentile, wavelet_LLL_firstorder_Maximum, squareroot_firstorder_Mean and exponential_firstorder_Median in PET modality (Figure 2C) and wavelet_LLL_glszm_LargeAreaHighGrayLevelEmphasis, original_glszm_LargeAreaHighGrayLevelEmphasis, square_glrIm_RunVariance, logarithm_firstorder_Skewness, square_glrIm_LongRunLowGrayLevelEmphasis, square_glrIm_LongRunEmphasis, logarithm_firstorder_Mean, square_gldm_LargeDependenceLowGrayLevelEmphasis, square_gldm_LargeDependenceEmphasis and original_glszm_ZoneVariance in CT modality (Figure 2D). In summary, your constructive suggestion helps establish the bridge between deep learning features and conventional texture features, which further enhances the interpretability of our DLNMS and improve the quality of our manuscript. Again, we thank for your thoughtful comments, which does inspire and benefit us a lot.

Changes in the text:

Result section Line 156-158: In addition, the DLMNS was represented by conventional PET and CT texture features in ONM prediction, implying the significant correlations between the DLNMS and PET/CT texture features (Fig. 2).

Fig. 2. PET and CT texture features related to the DLNMS.

(A) Top 10 PET and (B) top 10 CT texture features related to the DLNMS N1 prediction. (C)

Top 10 PET and (D) top 10 CT texture features related to the DLNMS N2 prediction.

DLNMS, deep learning nodal metastasis signature; PET, positron emission tomography; CT, computed tomography.

4. High-resolution CT findings is necessary to analyze the subtle images therefore it is difficult to obtain such precise findings using the conventional CT images associated with PET examination. Comments should be added.

Reply:

We thank the reviewer for raising this significant question. We agree with you that high-resolution CT findings are crucial to obtain such precise findings. Therefore, in this revision,

we have added this point as one of the limitations of this study in the discussion section.

Changes in the text:

Discussion section Line 407-410: Thirdly, high-resolution CT findings is necessary to analyze the subtle images, however, not all PET/CT equipment harbor the capability of outputting such high-quality images, which might reduce the clinical applicability of the DLNMS in certain institutions.

Reviewer 2:

1. Overall, the main text is written well, albeit quite dense, especially because many details are tucked away in Supplemental Materials and aren't easy to find (e.g. positive/negative ratios for train/val sets are only available in Figure S1). Despite format limitations, I still encourage the authors to make an effort in improving the text flow and make more specific references to supplemental materials.

Reply:

We appreciated the reviewer for pointing this out. The journal requires strict word limits. Therefore, some method details could only be provided in the supplementary material. We agree with you that making details easily to find would improve readability of this manuscript. In this revision, we have made positive/negative ratios for train/validation sets, algorithm structures and methods for multiple comparisons clear in the main text. In addition, we have added more specific references to supplemental materials.

Changes in the text:

Method section Line 420-428: The study design is described in Fig. 1. The DLNMS was developed using an internal cohort (entire: n=1911, occult N1 proportion=11.64%, occult N2 proportion=8.42%; training: n=1528, occult N1 proportion=11.45%, occult N2 proportion=8.31%; validation: n=383, occult N1 proportion=12.79%, occult N2 proportion=7.57%). Subsequently, a multicenter external cohort (n=355, occult N1 proportion=11.27%, occult N2 proportion=9.90%) and a multicenter prospective cohort (n=999,

occult N1 proportion=11.64%, occult N2 proportion=8.41%; ClinicalTrials.gov, NCT05425134) were adopted to fully validate the predictive efficiencies of the DLNMS by benchmarking the single-modal deep learning model, clinical model and physicians.

Method section Line 450-456: Two ResNet18 backbone networks with the same structure were used to extract features from PET and CT images separately. Then, the PET and CT features were fused using the concat operation and input into a fully connected layer for classifications of ONM. The DLNMS consisted of two separate models predicting occult N1 and N2, respectively. For occult N1 prediction, data were divided into N1 metastasis and non-N1 metastasis. Similarly, in N2 prediction, data were divided into N2 metastasis and non-N2 metastasis.

Method section Line 509-511: The Benjaminiand Hochberg method was utilized to correct p values from multiple comparisons in DeLong's tests and gene analyses.

Supplementary Method 13. References

- 1 Hariharan, B., Arbeláez, P., Girshick, R. & Malik, J. in Proceedings of the IEEE conference on computer vision and pattern recognition. 447-456.
- 2 Lee, K., Zung, J., Li, P., Jain, V. & Seung, H. S. Superhuman accuracy on the SNEMI3D

connectomics challenge. arXiv preprint arXiv:1706.00120 (2017).

3 Quan, T., Hildebrand, D. & Jeong, W. Fusionnet: A deep fully residual convolutional neural network for image segmentation in connectomics. arXiv 2016. arXiv preprint arXiv:1612.05360 (2016).

4 Beier, T. et al. Multicut brings automated neurite segmentation closer to human performance. Nature methods 14, 101-102 (2017).

5 Toubal, I. E., Duan, Y. & Yang, D. in 2020 IEEE Applied Imagery Pattern Recognition Workshop (AIPR). 1-9 (IEEE).

6 Zeng, T., Wu, B. & Ji, S. DeepEM3D: approaching human-level performance on 3D anisotropic EM image segmentation. Bioinformatics (Oxford, England) 33, 2555-2562 (2017).

7 Simonyan, K. & Zisserman, A. Very deep convolutional networks for large-scale image recognition. arXiv preprint arXiv:1409.1556 (2014).

8 He, K., Zhang, X., Ren, S. & Sun, J. in Proceedings of the IEEE conference on computer vision and pattern recognition. 770-778.

9 Russakovsky, O. et al. Imagenet large scale visual recognition challenge. International journal of computer vision 115, 211-252 (2015).

10 Goodfellow, I., Bengio, Y. & Courville, A. 6.2. 2.3 softmax units for multinoulli output distributions. Deep learning 180 (2016).

11 Brownlee, J. Probability for machine learning: Discover how to harness uncertainty with

Python. (Machine Learning Mastery, 2019).

12 Loshchilov, I. & Hutter, F. Sgdr: Stochastic gradient descent with warm restarts. arXiv preprint arXiv:1608.03983 (2016).

13 Loshchilov, I. & Hutter, F. Decoupled weight decay regularization. arXiv preprint arXiv:1711.05101 (2017).

14 Loshchilov, I. & Hutter, F. Fixing weight decay regularization in adam. (2018).

15 Zhang, H., Cisse, M., Dauphin, Y. N. & Lopez-Paz, D. mixup: Beyond empirical risk minimization. arXiv preprint arXiv:1710.09412 (2017).

16 Lin, T.-Y., Goyal, P., Girshick, R., He, K. & Dollár, P. in Proceedings of the IEEE international conference on computer vision. 2980-2988.

17 Huang, G., Liu, Z., Van Der Maaten, L. & Weinberger, K. Q. in Proceedings of the IEEE conference on computer vision and pattern recognition. 4700-4708.

2. However, the study could benefit from a more thorough comparison with existing literature and methods. The authors review established literature very briefly and selectively and do not describe the state-of-the-art, which leads to a poorly defined knowledge gap that this study is trying to bridge. Although there may be no prior applications of deep learning to specifically predict NSCLC ONM, there is rich literature on related topics, from using conventional image processing techniques to high-performing multimodal deep learning for related tasks on

PET/CT images (PMID: 36029345, PMID: 32773965, PMID: 33591922). E.g., see (PMID: 36870800) for overview of over a dozen studies that applied AI to PET/CT imaging data for management of lymphoma. Even if there is no previous method that DLNMS can be compared to, context for the general use of AI for similar tasks should be provided to establish the novelty and significance of the study in the context of the broader field.

Reply:

Thank you for pointing this out and we have acknowledged our inadequacy in literature review. As you commented, despite limited applications of PET/CT deep learning to specifically predict NSCLC ONM, there is rich literature on deep learning for related tasks on PET/CT images. Therefore, in the introduction section, we have described the general use of AI on PET/CT images to emphasize the novelty and significance of the study. Additionally, in the discussion section, we have summarized the multimodal algorithms in PET/CT analysing and clarify the reason why we choose feature-level fusion based on the ResNet 18 to construct our DLNMS.

Changes in the text:

Introduction section Line 95-112: The deep learning technology which allows the high-dimensional quantification of radiological images and greater extraction of detailed characterizations than the human vision, has been proposed as a revolutionary approach for disease diagnoses, prognosis evaluations, and therapeutic decisions. PET/CT, which is capable of capturing the anatomic and metabolic representations of tumors, has been

leveraged as a dependable imaging modality to characterize malignancy grade and metastasis burden. Its multimodal nature, on the one hand increases the feature dimensions and information abundance, but on the other hand poses a higher requirement for the deep learning algorithm.

With the development of multimodal algorithms, the current deep learning technology has evolved to be an effective method for PET/CT image analyzing, which harbors the capability of taking full advantages of the complementary information of PET and CT modalities. It has been demonstrated that multimodal deep learning algorithms shown potentials in cancer identification, tumor segmentation, and risk quantification based on PET/CT imaging. Despite these tremendous breakthroughs, the application of PET/CT based deep learning for ONM prediction of lung cancer is limited. We hypothesize that cross-modal dominance complementation based on PET and CT imaging is capable of quantifying ONM probability to support the comprehensive treatments of clinical N0 NSCLC, and the captured ONM risks would be associated with histologic, genetic, and microenvironment behaviors.

Discussion section Line 359-365: With the development of multimodal algorithms, the deep learning approach has been applied to analyze PET/CT imaging. Based on the main advancements of deep learning technology, multimodal fusion primarily involved three strategies: input-level concatenation, feature-level combination, and output-level average. Our preliminary experiments investigated multiple deep learning architectures and fusion strategies, revealing feature-level fusion based on the ResNet 18 backbone yield better efficiencies and was finally utilized to generate our DLNMS.

3. One issue that still needs to be addressed is that the study seems to lack proper multiple testing correction for the reported significance results. There are many tests performed that are used to compare prediction approaches, variables, etc. increasing the risk of false discoveries. The authors should make sure to apply appropriate correction methods to account for this issue and ensure the validity of their findings. The correction procedures should be clearly described in the Statistical analysis subsection.

Reply:

We appreciated the reviewer for this helpful comment. To compare predictive efficiency between DLNMS and other models, we performed multiple Delong's tests. In this manuscript, 5 comparisons (DLNMS versus PET model, DLNMS versus CT model, DLNMS versus clinical model, DLNMS versus senior physicians and DLNMS versus junior physicians) were simultaneously conducted. Therefore, the Benjamini and Hochberg corrections for 5 multiple comparisons were utilized to reduce the risk of false discoveries, results after correction procedures were provided in the Table-response 1, p value for multiple Delong's tests remained significant in all cohorts. In addition, in GSEA and ssGSEA analyses, the Benjamini and Hochberg corrections was also adopted to adjust p values. In this revision, we have clarified these in the statistic section.

Changes in the text:

Method section Line 509-511: The Benjaminiand Hochberg method was utilized to correct p values from multiple comparisons in Delong's tests and gene analyses.

Fig. 3. Predictive performances of the DLNMS for occult nodal metastasis in clinical stage N0 non-small cell lung cancer.

Fig. 4. Predictive performances of the DLNMS for occult N2 metastasis diagnosed by nodal biopsy in clinical stage N0 non-small cell lung cancer.

4. I have to point out that throughout the manuscript, the authors keep referring to various values of AUC as “satisfactory” or “unsatisfactory” predictive performances. Such assessment should be either substantiated by quantitative definition of what is “satisfactory” and why or should be avoided altogether.

Reply:

We appreciated this important comment and we agree with you that the term “satisfactory” was not been defined quantitatively, which might mislead readers. Therefore, in this revision, we have deleted these descriptions altogether.

5. Finally, while the authors did compare the performance of the DLNMS model with traditional clinical decision-making tools and human expert performance, it would be helpful to expand on these comparisons in the discussion section. This could involve exploring potential reasons for the observed differences in performance and discussing the implications of these findings for clinical practice.

Reply:

We thank the reviewer for raising this important question and we have acknowledged our inadequacy in discussion. There were many publications constructing traditional clinical decision-making tools to predict occult nodal metastasis. In this revision, for the second paragraph of the discussion section, we summarized publications regarding clinical physicians and clinical models for ONM predictions and speculated that potential reasons of poor

performance of clinical physicians are that this method is very subjective due to heterogenous experiences among physicians. In addition, we further compared clinical models to clinical physicians and speculated that potential reasons of improved performance of clinical models are that this method are capable of comprehensively estimating the probability of ONM to directly quantify the ONM probabilities of a given patient. Finally, we further discussed performances of these clinical models were far from meeting clinical requirements. We speculated that potential reasons are that more valuable radiographic features for predicting ONM are not been investigated by conventional clinical models. Therefore, the deep learning approach which quantifies more valuable radiological features is required and this promising method is further discussed in the third paragraph of the discussion section.

Changes in the text:

Discussion section Line 338-367: In clinical practice, clinical physicians mainly rely on certain clinical characteristics especially imaging features to capture the ONM risks of clinical stage N0 NSCLC. Evidences have emerged that metabolic and morphologic parameters on PET/CT, such as tumor size, central location, consolidation ratio, and metabolic value might provide efficient clues for ONM diseases. Nevertheless, this subjective evaluation yields low AUROCs of 0.525-0.676 due to heterogenous experiences among physicians, and is incapable of comprehensively estimating the probability of ONM, so as to convey a direct implication to the management strategy for a given patient. The triumph of individually

quantifying ONM risks based on predictive models represented a crucial step. Predictive rules integrating clinical variables could calculate the probability of ONM involvement in clinical N0 NSCLC. However, in spite of their higher accuracies than clinical physicians, these clinical models were far from meeting clinical requirements, resulting in AUROCs of 0.700-0.756, which was also observed by the current study, our clinical model only yielded AUROCs of 0.675-0.794 for ONM identification. As such, more valuable radiographic features for predicting ONM should be investigated to achieve clinical utility.

Radiomics, which allows quantitative extraction of high-dimensional radiological features, has provided a promising approach for more accurate evaluation of the lymph node status of lung cancer. Several studies have been successful in recognizing ONM in early-stage NSCLC utilizing radiomics phenotypes, which yielded AUROC values of 0.808 to 0.820. Despite such inspiring success, the above radiomics studies were limited in the CT modal, and the added value of PET radiomics features for ONM prediction of NSCLC are still ambiguous. With the development of multimodal algorithms, the deep learning approach has been applied to analyze PET/CT imaging. Based on the main advancements of deep learning technology, multimodal fusion primarily involved three strategies: input-level concatenation, feature-level combination, and output-level average. Our preliminary experiments investigated multiple deep learning architectures and fusion strategies, revealing feature-level fusion based on the ResNet 18 backbone yield better efficiencies and was finally utilized to generate our DLNMS. The current study demonstrated that the cross-modal DLNMS incorporating PET and CT radiomics features achieved AUROCs of 0.875-0.958, make it superior to single-modal models

based on PET or CT alone for ONM prediction.

6. The task setting is not clear. Data labeling and model training descriptions indicate that the model performs binary classification, however, separate scores are reported for N1 and N2 ONM (Figure 2A,B). Does that mean that 2 separate models were trained to predict N0 vs N1 and N0 vs N2? The authors should clarify data splits and model training very explicitly.

Reply:

We thank the reviewer for this important question. In clinical practice, N1 ONM and N2 ONM have their independent values for treatment decisions. As such, we must obtain both predictive scores to guide clinical decisions. As you commented, 2 separate models were trained to predict non-N1 metastasis vs N1 and non-N2 metastasis vs N2 so as to calculate separate scores. Therefore, in N1 prediction, data were divided into N1 metastasis and non-N1 metastasis. Similarly, in N2 prediction, data were divided into N2 metastasis and non-N2 metastasis. In this revision, we have clarified this explicitly.

Changes in the text:

Method section Line 450-456: Two ResNet18 backbone networks with the same structure were used to extract features from PET and CT images separately. Then, the PET and CT features were fused using the concat operation and input into a fully connected layer for classifications of ONM. The DLNMS consisted of two separate models predicting occult N1 and N2, respectively. For occult N1 prediction, data were divided into N1 metastasis and non-N1 metastasis. Similarly, in N2 prediction, data were divided into N2 metastasis and non-N2

metastasis.

7. Another issue that is not discussed enough is substantial class imbalance (1/9 if N1/N2 are considered separately), which the authors only briefly mention in data preprocessing. While the authors balance classes during training, it does affect prediction performance as demonstrated by PPV values (Figures 2, 3) and confusion matrices (Figure S3). When such class imbalance is present, AUC curves can be misleading and AUPR curves should be preferred (PMID: 25738806). I suggest the authors to either report AUPR curves next to AUC curves, or replace the latter with the former and move AUC curves to supplemental materials. Correspondingly, Results and/or Discussion should mention this issue, as in some cases the model predicts as many (Figure S3 E) or even +30% more false positives compared to true positives (Figure S3 3G).

Reply:

We thank the reviewer for this important suggestion. We agree with you that on imbalanced datasets, the PR curve does provide more information on model's discriminative ability for positive cases (ONM patients) because precision ($TP/[TP + FP]$) and recall ($TP/[TP + FN]$) are both indicators for positive predictive abilities [1, 2]. However, we would like to point out that different from conventional classification tasks which focuses more on positive cases, ONM recognition, which would pose a direct impact on treatment decisions of patients, emphasizes model's discriminative abilities for both positive and negative cases. For example,

if a patient predicted as negative by models actually is ONM disease (false negative prediction), this patient would directly lose the opportunity of receiving optimal treatment, which is a terrible result for his/her survival. In contrast, if a patient predicted as positive by models actually is healthy disease (false positive prediction), unnecessary treatments (such as invasive biopsy, adjuvant chemotherapy) would be administrated to him/her. In summary, for the task of ONM classification, we must comprehensively consider the model's discriminative ability for both positive (sensitivity and PPV) and negative cases (specificity and NPV) to avoid missed diagnosis (false negative) and meanwhile improve positive diagnostic accuracy.

What needs to be emphasized is that the PR curve only focuses on the efficiency to identify diseased cases (precision: $TP/[TP + FP]$ and recall: $TP/[TP + FN]$) but ignores those correctly predicted healthy cases [3]. The AUROC is a summary indicator comprehensively quantifying the positive (sensitivity) and negative (specificity) predictive capabilities [4]. In addition, according to a publication investigating the performances of ROC and PR curves in the domain of epidemic diseases, they recommended to use both the AUCROC and the AUPRC but to prefer the AUPRC as the prevalence below 5% [5]. Therefore, we select ROC curves to evaluate the predictive efficiency of models considering that ROC curves comprehensively quantified both positive (sensitivity) and negative (specificity) discriminative. In addition, the cutoff values of DLNMS were determined by Youden Index in which the sum of sensitivity and specificity of DLNMS were maximal. By utilizing this cutoff values, as you pointed, confusion matrices revealed that the model predicts as many false positives compared to true positives. This result could also be explained by the Youden Index, we select

a cutoff value to make the sum of sensitivity and specificity maximal, under the circumstance of high specificity and NPV (both more than 90%), the sensitivity and PPV would relatively decreased. Actually, based on the current data, if we set a high DLNMS score (such as 0.9) as the cutoff value, the false positive cases would significantly decrease and the PPV could even reached 95%. However, that would significantly decrease the specificity and NPV and makes more patients be predicted as false negative, which mean more patients would directly lose the opportunity of receiving optimal treatments. In summary, we select ROC curve to comprehensively evaluate the predictive efficiency of DLNMS and determine the cutoff values to avoid missed diagnosis (false negative) and meanwhile improve positive diagnostic accuracy.

For the above reasons, we did not delete ROC curves from the main text in this revision and we agree with that addition of AUPRC would provide more information on model's discriminative ability for positive cases. Therefore, we report AUPRCs as an addition. However, Figure 2 have contained too many panels and Reviewer 1 suggested we provide two additional figures in the main text. After balancing the importance of AUPRCs and content limit, we reported AUPRCs of different models in the main text as a new Table 3. For N1 prediction, the DLNMS achieved AUPRCs of 0.882, 0.853, and 0.871 in the validation set, external cohort and prospective cohort, respectively. For N2 prediction, the DLNMS achieved AUPRCs of 0.876, 0.849, 0.863, and 0.857 in the validation set, external cohort, prospective cohort, and biopsy cohort, respectively. The DLNMS achieved higher AUCPRCs than other models. As shown in the above results, AUPRCs were relatively lower than AUROCs, which might be

explained by the imbalanced data, in this domain the number of negative examples greatly exceeds the number of positive examples. Consequently, a small change in the number of false positives can lead to a large change in the false positive rate used in PR analysis [6]. Additionally, as you suggested, we have discussed this point in the discussion section.

References

[1] Liu, Z. & Bondell, H. D. Binormal Precision–Recall Curves for Optimal Classification of Imbalanced Data. *Statistics in Biosciences* 11, 141-161, doi:10.1007/s12561-019-09231-9 (2019).

[2] Saito, T. & Rehmsmeier, M. The precision-recall plot is more informative than the ROC plot when evaluating binary classifiers on imbalanced datasets. *PLoS one* 10, e0118432, doi:10.1371/journal.pone.0118432 (2015).

[3] Ozenne, B., Subtil, F. & Maucort-Boulch, D. The precision–recall curve overcame the optimism of the receiver operating characteristic curve in rare diseases. *Journal of Clinical Epidemiology* 68, 855-859, doi:https://doi.org/10.1016/j.jclinepi.2015.02.010 (2015).

[4] Fawcett, T. An introduction to ROC analysis. *Pattern Recognition Letters* 27, 861-874, doi:https://doi.org/10.1016/j.patrec.2005.10.010 (2006).

[5] Ozenne B, Subtil F, Maucort-Boulch D. The precision--recall curve overcame the optimism of the receiver operating characteristic curve in rare diseases. *J Clin Epidemiol.* 2015

Aug;68(8):855-9. doi: 10.1016/j.jclinepi.2015.02.010

[6] Williams CKI. The Effect of Class Imbalance on Precision-Recall Curves. *Neural Comput.*

2021 Apr 1;33(4):853-857. doi: 10.1162/neco_a_01362

Changes in the text:

Table 3. Areas under precision-recall curves of different models for occult N1 and N2 prediction

Models	Occult N1 prediction			Occult N2 prediction		
	Validation set	External cohort	Prospective cohort	Validation set	External cohort	Prospective cohort
DLNMS	0.882	0.853	0.871	0.876	0.849	0.863
PET model	0.756	0.731	0.748	0.753	0.710	0.741
CT model	0.779	0.751	0.764	0.765	0.746	0.755
Clinical model	0.656	0.612	0.627	0.694	0.635	0.648
Senior physicians	0.504	0.562	0.538	0.563	0.583	0.569
Junior physicians	0.582	0.590	0.599	0.514	0.555	0.534

DLNMS, deep learning nodal metastasis signature; PET, positron emission tomography; CT, computed tomography.

Result section Line 167-171: The areas under the precision-recall curve (AUPRC), sensitivity, specificity, positive predictive value (PPV), positive predictive value (NPV) and accuracy of the DLNMS for predicting occult N1 and N2 metastasis were 0.882, 0.898, 0.928, 0.647, 0.984 and 0.924, and 0.876, 0.897, 0.842, 0.317, 0.990, and 0.846, respectively.

Result section Line 179-181: In addition, the AUPRC, sensitivity, specificity, PPV, NPV and accuracy of the DLNMS for predicting occult N1 and N2 metastasis were 0.853, 0.700, 0.905 0.483, 0.960 and 0.882, and 0.849, 0.857, 0.813, 0.333, 0.981, and 0.817, respectively.

Result section Line 189-191: Additionally, the AUPRC, sensitivity, specificity, PPV, NPV and accuracy of the DLNMS for occult N1 and N2

prediction were 0.871, 0.793, 0.926, 0.586, 0.971 and 0.911, and 0.863, 0.833, 0.828, 0.308, 0.982, and 0.829, respectively.

Result section Line 237-238: The AUPRC, sensitivity, specificity, PPV, NPV and accuracy of the DLNMS were 0.857, 0.919, 0.699, 0.436, 0.971 and 0.743, respectively (Fig. 4A & Table 3)

Discussion section Line 368-379: In the domain of machine learning, one issue worth mentioning is the method for performance evaluation. On an imbalanced dataset with a low proportion of positive classifications, the PR curve might be more effective than the ROC curve in quantifying positive discriminative ability. However, what needs to be emphasized is that the PR curve only focuses on the efficiency to identify diseased cases but ignores those correctly predicted healthy cases. Different from conventional classification tasks, ONM recognition would pose a direct impact on treatment decisions, which emphasizes model's discriminative abilities for both positive and negative subjects. If a patient diagnosed as healthy actually is ONM disease (false negative), this patient would directly lose the opportunity of receiving optimal treatments. The AUPRC is a summary indicator comprehensively quantifying the positive and negative predictive capabilities, we therefore chose the Youden Index based on ROC curves to determine the cutoff values of DLNMS.

8. *The main contribution of this study is assessment of the proposed model. However, any graphical or textual description of the model is missing from the main text. The authors should at least add a brief description of key characteristics to the main text and a diagram as a supplemental figure.*

Reply:

Thank you for pointing this out. In this study, two ResNet18 backbone networks with the same structure are used to extract features from PET and CT images separately. Then, the PET and CT features were fused using the concat operation and input into a fully connected layer for the classification of N1 and N2 nodules. In this revision, we have described key characteristics of our model and provided a diagram for model description as a supplemental figure.

Changes in the text:

Method section Line 450-456: Two ResNet18 backbone networks with the same structure were used to extract features from PET and CT images separately. Then, the PET and CT features were fused using the concat operation and input into a fully connected layer for classifications of ONM. The DLNMS consisted of two separate models predicting occult N1 and N2, respectively. For occult N1 prediction, data were divided into N1 metastasis and non-N1 metastasis. Similarly, in N2 prediction, data were divided into N2 metastasis and non-N2 metastasis.

Supplementary Fig. 8. Diagram illustrating the structure of DLNMS

9. *The choice of many hyperparameters for constructing and training the model is not justified. For example, choosing ResNet-18 as the base architecture for the DLNMS model may be suboptimal, considering that it is relatively shallow and outdated compared to more recent deep learning architectures. Did authors consider alternative models with superior performance in various image recognition tasks, e.g. deeper ResNet, EfficientNet, or DenseNet?*

Reply:

We appreciate this comment. Before choosing the final algorithm, we have tried several deep learning architectures including Resnet18, Resnet50, Resnet152, and DenseNet121. According to results of preliminary experiments, deeper ResNets were easily overfitting than Resnet18. In test set, the Resnet 50 achieved AUCs of 0.81-0.87 and 0.79-0.85 for N1 and N2 prediction, respectively. The Resnet 152 achieved AUCs of 0.72-0.79 and 0.74-0.78 for N1 and N2 prediction, respectively. Therefore, in ResNet architectures, we chosen Resnet 18. Subsequently, we further tried DenseNet121, DenseNet121 achieved AUCs of 0.83-0.89 and 0.84-0.91 for N1 and N2 prediction in testing. In addition, considering EfficientNet is more applied in mobile devices with limited computing power, we did not attempt this architecture in the primary experiments. However, as you mentioned, we further tried EfficientNet during revision, EfficientNet yield lower performance than ResNet 18, achieving AUCs of 0.79 to 0.86 and 0.74-0.84 for occult N1 and N2, respectively. The performances of all architectures were shown in Table-response 1.

In summary, the preliminary experiments revealed ResNet 18 achieved better efficiency and we finally chose ResNet 18 as the backbone of the DLNMS. We discussed above results and speculated that some deeper and more complex deep learning architectures might target on millions or even billions of data and thousand classification tasks, the current study only involved few thousand data and ONM prediction is a simple binary classification. Therefore, deeper and complex

architectures might be easier to overfit. In addition, ResNet 18 has been widely applied in medical image analyses [1-6] and we think ResNet 18 was well qualified for the medical task in this study.

References

- [1] Ozaki K, Kurose Y, Kawai K, Kobayashi H, Itabashi M, Hashiguchi Y, Miura T, Shiomi A, Harada T, Ajioka Y. Development of a diagnostic artificial intelligence tool for lateral lymph node metastasis in advanced rectal cancer. *Dis Colon Rectum*. 2023 Jun 1. doi: 10.1097/DCR.0000000000002719
- [2] Lee DK, Kim JH, Oh J, Kim TH, Yoon MS, Im DJ, Chung JH, Byun H. Detection of acute thoracic aortic dissection based on plain chest radiography and a residual neural network (Resnet). *Sci Rep*. 2022 Dec 19;12(1):21884. doi: 10.1038/s41598-022-26486-3
- [3] Liao H, Xu Y, Meng Q, Mao Z, Qiao Y, Liu Y, Zheng Q. A convolutional neural network-based, quantitative complete blood count scattergram-mapping framework promptly screens acute promyelocytic leukemia with high sensitivity. *Cancer*. 2023 May 31. doi: 10.1002/cncr.34890
- [4] Xu F, Xiong Y, Ye G, Liang Y, Guo W, Deng Q, Wu L, Jia W, Wu D, Chen S, Liang Z, Zeng X. Deep learning-based artificial intelligence model for classification of vertebral compression fractures: A multicenter diagnostic study. *Front Endocrinol (Lausanne)*. 2023 Mar 22;14:1025749. doi: 10.3389/fendo.2023.1025749
- [5] Lu SY, Wang SH, Zhang YD. SAFNet: A deep spatial attention network with classifier fusion for breast cancer detection. *Comput Biol Med*. 2022 Sep;148:105812. doi: 10.1016/j.combiomed.2022.105812
- [6] Liao H, Yang J, Li Y, Liang H, Ye J, Liu Y. One 3D VOI-based deep learning radiomics strategy, clinical model and radiologists for predicting lymph node metastases in pancreatic ductal adenocarcinoma based on multiphase contrast-

enhanced computer tomography. Front Oncol. 2022 Sep 9;12:990156. doi: 10.3389/fonc.2022.990156

Table-response 1: Performance of different deep learning architectures in testing set according to preliminary experiments

AUC	ResNet 50	ResNet 152	DenseNet 121	EfficientNet
Occult N1	0.81-0.87	0.72-0.79	0.83-0.89	0.79-0.86
Occult N2	0.79-0.85	0.74-0.78	0.84-0.91	0.74-0.84

Changes in the text:

Discussion section Line 359-365: With the development of multimodal algorithms, the deep learning approach has been applied to analyze PET/CT imaging. Based on the main advancements of deep learning technology, multimodal fusion primarily involved three strategies: input-level concatenation, feature-level combination, and output-level average. Our preliminary experiments investigated multiple deep learning architectures and fusion strategies, revealing feature-level fusion based on the ResNet 18 backbone yield better efficiencies and was finally utilized to generate our DLNMS.

Supplementary Method 7. Selection for deep learning architecture, fusion strategy, and image augmentation

To select the optimal deep learning architecture, preliminary experiments were performed between ResNet-18, ResNet-50, ResNet-152, and DenseNet-121, revealing that deeper ResNets were easily overfitting than ResNet-18 and DenseNet-121 achieved a slightly lower performance than ResNet-18. Therefore, ResNet-18 was chosen as the basis of the DLNMS. For the fusion strategies of PET and CT images, the following three methods: input-level concatenation, feature-level combination, and output-level average were investigated, and feature-level combination achieved the best performance and was therefore applied in the final DLNMS. For image augmentations, preliminary experiments were conducted to investigate the addition of a certain augmentation method would improve model performances, we primarily attempted several augmentation methods, revealing that the current augmentation flow consisted of random rotation, random shift, random crop, three orthogonal planes extraction, random sharpness, and random blur could increase the effective size of training data and alleviate the overfitting problem. In addition, transforms generated from the current augmentations could improve the generalization ability, which was beneficial for external validations.

10. *There are multiple important details missing from the description of the model in Supplemental Materials: (1) It's not clear how the multimodal PET/CT model was constructed, i.e. were two networks trained together with the final classifier, or was it just a combination of features from the two single-modality extractor (which is not truly multimodal)? Did the authors consider other options such as multi-task learning? (2) It should be explicitly stated that ResNet-18 the authors employed was pretrained on the ImageNet dataset of natural images. (3) There is no information on how image augmentations were selected, did the authors perform any ablation studies to make sure selected transforms do not hurt the performance? (4) There is no analysis of model computational requirements or convergence. (5) Supplementary Materials mention that the authors used binary cross-entropy as a loss function, while the code repository employs Focal loss instead.*

Reply:

Thank you for these important comments. Regarding the first question, we speculate you mean the issues of the deep fusion and late fusion. In preliminary experiments, we have tried two fusion methods. The first one is deep fusion (Figure-response 1), we combined features from the two single-modality and input them into the final fully connected layers. The second one is the late fusion (Figure-response 2), we speculate you mean this method. In this method, two networks trained together for each modality, and then the two single classifiers were combined to calculate the final results. In testing set, this method only achieved AUCs of 0.73-0.81 for N1 and N2 prediction. We speculate that the late fusion only combined outputs from two single-modality, these two outputs might contain redundant information, which did not take the advantage of end-and-end learning. In contrast, the deep fusion could extract features from two modalities and combine them in the feature layer, which could reduce overfitting risks and maximize the utilization of multimodal imaging features. Therefore,

we choose the deep fusion method. Actually, we also tried the early fusion method (Figure-response 3), in which PET and CT modalities were fused before inputting into the network. However, this method achieved the poorest performances compared to the deep fusion and late fusion methods, with AUC of 0.71-0.78 for N1 and N2 prediction. The early fusion, deep fusion and late fusion were different methods for multimodal learning, which have been applied in multiple classification tasks [1-5]. As you commented, the best results could only be achieved after trying different methods, compared to others, the deep fusion in the current task was more efficient, we therefore chose this method in this study. In addition, multi-task learning is the first option when we starting this study because it can solve two classifications using one model. In our preliminary experiments, however, multi-task learning did not achieve good performances with AUCs of 0.67-0.72 for N1 and N2 prediction and therefore discard this method. We speculated that the difficulty of data balancing might limit the performance of multi-task learning considering N1 and N2 proportion were both low in this study. Finally, considering the current study focuses more on clinical application rather than algorithm comparisons, we therefore did not describe much about algorithm selection procedures.

Regarding the second question, we did utilize the pretrained ResNet-18 because the pretrained ResNet-18 was beneficial for convergence and reducing training time when finetuning our model. This method has been applied by many studies regarding imaging analyses [6-9]. In this revision, we have clarified this point in the method section.

Regarding the third question, preliminary experiments were conducted to investigate the addition of a certain augmentation method would improve model performances, we primarily attempted several augmentation methods, revealing that the current augmentation flow consisted of random rotation, random shift, random crop, three orthogonal planes extraction, random sharpness, and random blur could

increase the effective size of training data and alleviated the overfitting problem. In addition, transforms generated from the current augmentations make the model learn more new cases, which could further improve the generalization ability of the model and benefit external validations. Data augmentations have been applied by multiple lung cancer imaging analyses [10-11]. In this revision, we have clarified this in the method section.

Regarding the fourth question, Over the training iteration, the loss in the training set decreased consistently and plateaued finally. In addition, the performance metrics in the validation set stabilized or shown only minor fluctuations. No overfitting or underfitting was observed in training process. Above results indicated the DLNMS reached convergence. To evaluate the model computational requirements, parameters and floating point operations (FLOPs) were calculated, the number of parameters and FLOPs for the DLNMS were 24.47M and 974.88M Mac. This point has been detailed in the method section in this revision.

Regarding the fifth question, we apologize for this terrible mistake, considering the problem of substantial class imbalance, we finally use focal loss to replace cross-entropy as the loss function. We did not modify the description in the supplementary files and we apologize for our carelessness. In this revision, we have clarified this point.

References

- [1] Huang, K., Shi, B., Li, X., Li, X., Huang, S., & Li, Y. (2022). Multi-modal sensor fusion for auto driving perception: A survey. arXiv preprint arXiv:2202.02703
- [2] Predicting treatment response from longitudinal images using multi-task deep learning. *Nature communications* 12, 1851, doi:10.1038/s41467-021-22188-y (2021).
- [3] Li, K., Zhang, R. & Cai, W. Deep learning convolutional neural network (DLCNN): unleashing the potential of (18)F-FDG PET/CT in lymphoma. *Am J Nucl Med Mol Imaging* 11, 327-331 (2021).

- [4] Kumar, A., Fulham, M., Feng, D. & Kim, J. Co-Learning Feature Fusion Maps from PET-CT Images of Lung Cancer. *IEEE Trans Med Imaging*, doi:10.1109/tmi.2019.2923601 (2019).
- [5] Donahue, J. et al. Long-Term Recurrent Convolutional Networks for Visual Recognition and Description. *IEEE Trans Pattern Anal Mach Intell* 39, 677-691, doi:10.1109/tpami.2016.2599174 (2017).
- [6] Zhang X, Xie W, Li Y, Jiang K, Fang L. REAF: Remembering Enhancement and Entropy-Based Asymptotic Forgetting for Filter Pruning. *IEEE Trans Image Process*. 2023;32:3912-3923. doi: 10.1109/TIP.2023.3288986
- [7] Hille G, Agrawal S, Tummala P, Wybranski C, Pech M, Surov A, Saalfeld S. Joint liver and hepatic lesion segmentation in MRI using a hybrid CNN with transformer layers. *Comput Methods Programs Biomed*. 2023 Jun 7;240:107647. doi: 10.1016/j.cmpb.2023.107647
- [8] Wang X, Huang Y, Zeng D, Qi GJ. CaCo: Both Positive and Negative Samples are Directly Learnable via Cooperative-adversarial Contrastive Learning. *IEEE Trans Pattern Anal Mach Intell*. 2023 Mar 28;PP. doi: 10.1109/TPAMI.2023.3262608
- [9] Chang X, Wang J, Zhang G, Yang M, Xi Y, Xi C, Chen G, Nie X, Meng B, Quan X. Predicting colorectal cancer microsatellite instability with a self-attention-enabled convolutional neural network. *Cell Rep Med*. 2023 Feb 21;4(2):100914. doi: 10.1016/j.xcrm.2022.100914
- [10] Zhao G, Feng Q, Chen C, Zhou Z, Yu Y. Diagnose Like a Radiologist: Hybrid Neuro-Probabilistic Reasoning for Attribute-Based Medical Image Diagnosis. *IEEE Trans Pattern Anal Mach Intell*. 2022 Nov;44(11):7400-7416. doi: 10.1109/TPAMI.2021.3130759
- [11] Xie Y, Xia Y, Zhang J, Song Y, Feng D, Fulham M, Cai W. Knowledge-based Collaborative Deep Learning for Benign-Malignant Lung Nodule Classification on

Figure-response 1: Diagram for the deep fusion method

Figure-response 2: Diagram for the late fusion method

Figure-response 3: Diagram for the early fusion method

Changes in the text:

Method section Line 450-456: Two ResNet18 backbone networks with the same structure were used to extract features from PET and CT images separately. Then, the PET and CT features were fused using the concat operation and input into a fully

connected layer for classifications of ONM. The DLNMS consisted of two separate models predicting occult N1 and N2, respectively. For occult N1 prediction, data were divided into N1 metastasis and non-N1 metastasis. Similarly, in N2 prediction, data were divided into N2 metastasis and non-N2 metastasis.

Supplementary Method 6. Neural network training process

Two separate models were trained to predict N1 and N2 metastasis so as to calculate separate scores. In N1 prediction, data were divided into N1 metastasis and non-N1 metastasis. Similarly, in N2 prediction, data were divided into N2 metastasis and non-N2 metastasis. In order to obtain the occult N1/N2 metastasis probability, we constructed a neural network model based on the ResNet-18 algorithm. The model consisted of two parts: the feature extractor and the feature classifier. The input of the feature extractor was a tensor of size $3 \times 112 \times 112$, and the main structure of this part was ResNet-18 which contains 8 residual blocks. The pretrained ResNet-18 based on ImageNet dataset was utilized, which enabled better parameter initialization for medical image analysis tasks. The latter part classified the output map of the feature extractor and output two prediction probabilities, which represented the probability of metastasis and the probability of non-metastasis (sum to 1). The classifiers consisted of two fully-connected layers, the first layer had 512 output nodes, and the second one contained two nodes whose output values were calculated by the softmax function and converted into probabilities.

Additionally, for the deep learning nodal metastasis signature (DLNMS), we used two ResNet-18's feature extractors to extract the features from PET and CT, respectively. Then we combined PET and CT features as the following classifier's input. Models were trained with a mini-batch size of 32. We started training with a learning rate of 0.001 using CosineAnnealingWarmRestarts learning rate scheduler, and the restart epoch was set at 50. The entire network was trained by the adamw optimizer for 100

epochs. We used the MixUp algorithm as an additional data augmentation method. For the loss function, we used the binary focal loss function. The code implementation was based on the pytorch. The training of the algorithm was performed on a computer with a NVIDIA 3090.

Over the training iteration, the loss in the training set decreased consistently and plateaued finally. In addition, the performance metrics in the validation set stabilized and shown only minor fluctuations. No overfitting or underfitting was observed in the training process. Above results indicated the DLNMS reached convergence. To evaluate the model computational requirements, parameters and floating point operations (FLOPs) were calculated, the number of parameters and FLOPs for the DLNMS were 24.47M and 974.88M Mac, respectively.

Supplementary Method 7. Selection for deep learning architecture, fusion strategy, and image augmentation

To select the optimal deep learning architecture, preliminary experiments were performed between ResNet-18, ResNet-50, ResNet-152, and DenseNet-121, revealing that deeper ResNets were easily overfitting than ResNet-18 and DenseNet-121 achieved a slightly lower performance than ResNet-18. Therefore, ResNet-18 was chosen as the basis of the DLNMS. For the fusion strategies of PET and CT images, the following three methods: input-level concatenation, feature-level combination, and output-level average were investigated, and feature-level combination achieved the best performance and was therefore applied in the final DLNMS. For image augmentations, preliminary experiments were conducted to investigate the addition of a certain augmentation method would improve model performances, we primarily attempted several augmentation methods, revealing that the current augmentation

flow consisted of random rotation, random shift, random crop, three orthogonal planes extraction, random sharpness, and random blur could increase the effective size of training data and alleviate the overfitting problem. In addition, transforms generated from the current augmentations could improve the generalization ability, which was beneficial for external validations.

Supplementary Fig. 8. Diagram illustrating the structure of DLNMS

11. *Methods section need to be updated according to critiques above in order to allow other researchers to reproduce the method. Nevertheless, the authors could further enhance reproducibility by providing pre-trained models as supplementary material.*

Reply:

Thank you for this comment and we agree that more details regarding method is potentially important for reproducing. In this revision, we have updated the method section according to your comments. In addition, we have uploaded the pre-trained model at <https://github.com/zhongthoracic/DLNMS>.

12. *square brackets are missing around some confidence intervals*

Reply:

Thank you for this comment and we have revised it throughout the paper.

Changes in the text:

Result section Line 138-243:

Variables associated with ONM

As displayed in Table 2, in the training set, a younger age (odds ratio [OR]: 0.967, 95% confidence interval [CI]: [0.951, 0.984], $p < 0.001$), male sex (OR: 1.403, 95% CI: [1.001, 1.990], $p = 0.047$), pure solid type (OR: 2.525, 95% CI: [1.638, 3.891], $p < 0.001$), left location (OR: 1.512, 95% CI: [1.088, 2.100], $p = 0.014$), central location (OR: 1.743, 95% CI: [1.202, 2.530], $p = 0.003$) and larger tumor size (OR: 1.146, 95% CI: [1.001, 1.313], $p = 0.049$) were identified as independent predictors for occult N1 metastasis, and the pure solid type (OR: 3.389, 95% CI: [1.999, 5.745], $p < 0.001$) and higher SUVmax (OR: 1.144, 95% CI: [1.084, 1.330], $p = 0.040$) were independently related to occult N2 involvement. Most variables remained predictive for patients in the validation set, external cohort and prospective cohort (Supplementary Table 1). In addition, after incorporation of the DLNMS into analyses (Supplementary Table 2 & 3), the DLNMS was revealed as independent predictors for both occult N1 and N2 involvements.

Predictive performance of DLNMS

With an increase of DLNMS scores, more cases with occult N1 and N2 tumors were observed in the training set (Supplementary Fig. 1A & B), validation set (Supplementary Fig. 1C & D), external cohort (Supplementary Fig. 1E & F) and prospective cohort (Supplementary Fig. 1G & H). In addition, the DLNMS was represented by conventional PET and CT texture features in ONM prediction, implying the significant correlations between the DLNMS and PET/CT texture features (Fig. 2).

As illustrated in Fig. 3A & B, Table 3 and Supplementary Fig. 2, in the validation set, the abilities of the DLNMS to predict occult N1 and N2 diseases were shown to have areas under the receive operating characteristic curve (AUROCs) of 0.958 (95% CI: [0.923, 0.992]) and 0.942 (95% CI: [0.911, 0.973]), respectively, which were

significantly better than 0.873 (95% CI: [0.835, 0.911]) and 0.761 (95% CI: [0.680, 0.842]) of the PET model, 0.913 (95% CI: [0.875, 0.952]) and 0.887 (95% CI: [0.823, 0.952]) of the CT model, 0.752 (95% CI: [0.685, 0.819]) and 0.690 (95% CI: [0.603, 0.776]) of the clinical model, 0.612 (95% CI: [0.536, 0.689]) and 0.672 (95% CI: [0.574, 0.771]) of the senior physicians, and 0.616 (95% CI: [0.544, 0.687]) and 0.556 (95% CI: [0.465, 0.647]) of the junior physicians (DeLong's test: all $p < 0.05$). The areas under the precision-recall curve (AUPRC), sensitivity, specificity, positive predictive value (PPV), positive predictive value (NPV) and accuracy of the DLNMS for predicting occult N1 and N2 metastasis were 0.882, 0.898, 0.928, 0.647, 0.984 and 0.924, and 0.876, 0.897, 0.842, 0.317, 0.990, and 0.846, respectively.

In the external cohort (Fig. 3C & D), the DLNMS achieved AUROCs of 0.879 (95% CI: [0.813, 0.946]) and 0.875 (95% CI: [0.820, 0.930]) in predicting occult N1 and N2 metastasis, respectively, and were significantly superior than the PET model (0.790, 95% CI: [0.733, 0.847] and 0.727, 95% CI: [0.649, 0.805]), the CT model (0.826, 95% CI: [0.747, 0.905] and 0.817, 95% CI: [0.748, 0.887]), the clinical model (0.722, 95% CI: [0.642, 0.802] and 0.723, 95% CI: [0.648, 0.797]), the senior physicians (0.676, 95% CI: [0.590, 0.763] and 0.645, 95% CI: [0.554, 0.735]), and the junior physicians (0.633, 95% CI: [0.548, 0.719] and 0.594, 95% CI: [0.503, 0.685]) (DeLong's test: all $p < 0.05$).

In addition, the AUPRC, sensitivity, specificity, PPV, NPV and accuracy of the DLNMS for predicting occult N1 and N2 metastasis were 0.853, 0.700, 0.905, 0.483, 0.960 and 0.882, and 0.849, 0.857, 0.813, 0.333, 0.981, and 0.817, respectively.

In the prospective cohort (Fig. 3E & F), the DLNMS achieved AUROCs of 0.914 (95% CI: [0.877, 0.949]) and 0.919 (95% CI: [0.886, 0.942]) in discriminating occult N1 and N2 involvements, and were evidently better than the PET model (0.796, 95% CI: [0.751, 0.841] and 0.712, 95% CI: [0.656, 0.768]), the CT model (0.828, 95% CI: [0.777, 0.879] and 0.835, 95% CI: [0.779, 0.891]), the clinical model (0.794, 95% CI: [0.708, 0.791] and 0.675, 95% CI: [0.629, 0.721]), the senior physicians (0.672, 95% CI: [0.623, 0.722])

and 0.670, 95% CI: [0.613, 0.723]), and the junior physicians (0.645, 95% CI: [0.596, 0.693] and 0.635, 95% CI: [0.580, 0.691]) (DeLong's test: all $p < 0.05$). Additionally, the AUPRC, sensitivity, specificity, PPV, NPV and accuracy of the DLNMS for occult N1 and N2 prediction were 0.871, 0.793, 0.926, 0.586, 0.971 and 0.911, and 0.863, 0.833, 0.828, 0.308, 0.982, and 0.829, respectively.

In subgroup analyses regarding pathological types for patients in the validation set, external cohort and prospective cohort, the DLNMS achieved AUROCs of 0.916 (95% CI: [0.885, 0.947]) and 0.934 (95% CI: [0.915, 0.953]) in adenocarcinoma population for occult N1 and N2 prediction, respectively. Additionally, for squamous cell carcinoma population, the DLNMS yielded AUROCs of 0.904 (95% CI: [0.842, 0.966]) and 0.858 (95% CI: [0.779, 0.937]) for occult N1 and N2 prediction, respectively (Fig. 3G & H).

For patients in the validation set, external cohort and prospective cohort, the DLNMS could correct 38.30% occult N1, 73.11% benign N1, 78.13% occult N2, and 53.04% benign N2 diseases in those incorrectly diagnosed by the PET model (Supplementary Fig. 3A & B). Similarly, for those incorrectly predicted by the CT model, the DLNMS could correct 35.42% occult N1, 67.06% benign N1, 93.80% occult N2, and 41.18% benign N2 diseases (Supplementary Fig. 3C & D).

The calibration curves revealed that the DLNMS yielded better performances than others (Supplementary Fig. 4). Furthermore, we evaluated the clinical usefulness of the DLNMS compared to single-modal models for ONM detection via decision curve analyses, indicating that the DLNMS achieved better net benefits than other models no matter for occult N1 or N2 prediction (Supplementary Fig. 5). As summarized in Supplementary Table 4, the positive values of integrated discrimination improvements (all $p < 0.05$) and net reclassification index (all $p < 0.05$) for occult N1 and N2 predictions could be achieved when comparing the DLNMS to single-modal models.

Decision support for nodal biopsy

For 366 patients receiving nodal biopsy (Supplementary Table 5), the DLNMS yielded an AUROC of 0.853 (95% CI: [0.812, 0.895]) for predicting occult N2 diseases, which was significantly better than the PET model (0.644, 95% CI: [0.573, 0.715]), the CT model (0.780, 95% CI: [0.718, 0.841]), the clinical model (0.543, 95% CI: [0.471, 0.715]), the senior physicians (0.621, 95% CI: [0.554, 0.688]), and the junior physicians (0.525, 95% CI: [0.457, 0.594]). The AUPRC, sensitivity, specificity, PPV, NPV and accuracy of the DLNMS were 0.857, 0.919, 0.699, 0.436, 0.971 and 0.743, respectively (Fig. 4A & Table 3). In addition, with an increase in the DLNMS scores, more patients with occult N2 tumors were observed in the nodal biopsy cohort (Fig. 4B). Moreover, the DLNMS could correct 79.13% occult N2 and 56.41% benign N2 diseases in patients incorrectly diagnosed by the PET model (Fig. 4C). Similarly, for those incorrectly predicted by the CT model, the DLNMS could correct 100% occult N2 and 41.50% benign N2 diseases (Fig. 4D).

13. please add subtitles in Figure 2, panels G-J; Figure 3, panels A-D

Reply:

Thank you for this suggestion. In this revision, considering the content limit, the old Figure 2G-J have been moved to the new Supplementary Fig. 3. and the old Figure 3 has been moved to new Fig. 4. Subtitles of new Supplementary Fig. 3 and new Fig. 4A-D have been added.

Changes in the text:

Supplementary Fig. 3. Scatter graphs illustrating the DLNMS correct cases falsely predicted by the (A & B) PET and (C & D) CT models in the validation set, external cohort and prospective cohort.

Fig. 4. Predictive performances of the DLNMS for occult N2 metastasis diagnosed by nodal biopsy in clinical stage N0 non-small cell lung cancer.

14. please proofread *Supplementary Information*, there are many typos (e.g., line 74: retrospective -> retrospectively)

Reply:

Thank you for this comment and we apologize for our mistake. In this revision, we have checked spellings throughout the paper.

Changes in the text:

Supplementary Method 5. Data pre-processing process

The data pre-processing process was divided into ten steps: (1) Resampled each voxel in the original images to $0.6 \times 0.6 \times 0.6$ mm³ in the spatial dimension; (2) ROI of the lesion was randomly rotated $-180 \sim 180$ around its center; (3) Extracted the 3D ROI with a ROI size of $150 \times 150 \times 150$ voxels; (4) Randomly shifted $-3 \sim 3$ pixels lesion location as the center; (5) Extracted the three orthogonal scanning planes through the center of the lesion and stack them as pseudo-RGB images with size of $3 \times 150 \times 150$; (6) Randomly cropped the stacks to 132×132 ; (7) Randomly cropped and resize the stacks to 112×112 ; (8) Randomly sharpen the stacks; (9) Randomly blurred the stacks; (10) Subtracted the mean and divide the variance. In addition, since the number of negative samples was much more than the number of positive samples in the dataset, we randomly up-sampled the positive ones at the beginning of each training epoch so that the ratio of positive to negative samples was kept at 1:1.

Supplementary Method 6. Neural network training process

Two separate models were trained to predict N1 and N2 metastasis so as to calculate separate scores. In N1 prediction, data were divided into N1 metastasis and non-N1 metastasis. Similarly, in N2 prediction, data were divided into N2 metastasis and non-N2 metastasis. In order to obtain the occult N1/N2 metastasis probability, we constructed a neural network model based on the ResNet-18 algorithm. The model

consisted of two parts: the feature extractor and the feature classifier. The input of the feature extractor was a tensor of size $3 \times 112 \times 112$, and the main structure of this part was ResNet-18 which contains 8 residual blocks. The pretrained ResNet-18 based on ImageNet dataset was utilized, which enabled better parameter initialization for medical image analysis tasks. The latter part classified the output map of the feature extractor and output two prediction probabilities, which represented the probability of metastasis and the probability of non-metastasis (sum to 1). The classifiers consisted of two fully-connected layers, the first layer had 512 output nodes, and the second one contained two nodes whose output values were calculated by the softmax function and converted into probabilities.

Additionally, for the deep learning nodal metastasis signature (DLNMS), we used two ResNet-18's feature extractors to extract the features from PET and CT, respectively. Then we combined PET and CT features as the following classifier's input. Models were trained with a mini-batch size of 32. We started training with a learning rate of 0.001 using CosineAnnealingWarmRestarts learning rate scheduler, and the restart epoch was set at 50. The entire network was trained by the adamw optimizer for 100 epochs. We used the MixUp algorithm as an additional data augmentation method. For the loss function, we used the binary focal loss function. The code implementation was based on the pytorch. The training of the algorithm was performed on a computer with a NVIDIA 3090.

Over the training iteration, the loss in the training set decreased consistently and plateaued finally. In addition, the performance metrics in the validation set stabilized and shown only minor fluctuations. No overfitting or underfitting was observed in the training process. Above results indicated the DLNMS reached convergence. To evaluate the model computational requirements, parameters and floating point

operations (FLOPs) were calculated, the number of parameters and FLOPs for the DLNMS were 24.47M and 974.88M Mac, respectively.

Supplementary Method 7. Selection for deep learning architecture, fusion strategy, and image augmentation

To select the optimal deep learning architecture, preliminary experiments were performed between ResNet-18, ResNet-50, ResNet-152, and DenseNet-121, revealing that deeper ResNets were easily overfitting than ResNet-18 and DenseNet-121 achieved a slightly lower performance than ResNet-18. Therefore, ResNet-18 was chosen as the basis of the DLNMS. For the fusion strategies of PET and CT images, the following three methods: input-level concatenation, feature-level combination, and output-level average were investigated, and feature-level combination achieved the best performance and was therefore applied in the final DLNMS. For image augmentations, preliminary experiments were conducted to investigate the addition of a certain augmentation method would improve model performances, we primarily attempted several augmentation methods, revealing that the current augmentation flow consisted of random rotation, random shift, random crop, three orthogonal planes extraction, random sharpness, and random blur could increase the effective size of training data and alleviate the overfitting problem. In addition, transforms generated from the current augmentations could improve the generalization ability, which was beneficial for external validations.

15. Supplementary Information should have a separate References section and cite methods and packages used to perform the analyses (e.g. ResNet-18, ImageNet, etc.)

Reply:

Thank you for this comment. In this revision, we have added a separate references

section to cite methods ad packages used to perform the analyses.

Changes in the text:

Supplementary Method 13. References

- 1 Hariharan, B., Arbeláez, P., Girshick, R. & Malik, J. in Proceedings of the IEEE conference on computer vision and pattern recognition. 447-456.
- 2 Lee, K., Zung, J., Li, P., Jain, V. & Seung, H. S. Superhuman accuracy on the SNEMI3D connectomics challenge. arXiv preprint arXiv:1706.00120 (2017).
- 3 Quan, T., Hildebrand, D. & Jeong, W. Fusionnet: A deep fully residual convolutional neural network for image segmentation in connectomics. arXiv 2016. arXiv preprint arXiv:1612.05360 (2016).
- 4 Beier, T. et al. Multicut brings automated neurite segmentation closer to human performance. Nature methods 14, 101-102 (2017).
- 5 Toubal, I. E., Duan, Y. & Yang, D. in 2020 IEEE Applied Imagery Pattern Recognition Workshop (AIPR). 1-9 (IEEE).
- 6 Zeng, T., Wu, B. & Ji, S. DeepEM3D: approaching human-level performance on 3D anisotropic EM image segmentation. Bioinformatics (Oxford, England) 33, 2555-2562 (2017).
- 7 Simonyan, K. & Zisserman, A. Very deep convolutional networks for large-scale image recognition. arXiv preprint arXiv:1409.1556 (2014).
- 8 He, K., Zhang, X., Ren, S. & Sun, J. in Proceedings of the IEEE conference on computer vision and pattern recognition. 770-778.
- 9 Russakovsky, O. et al. Imagenet large scale visual recognition challenge. International journal of computer vision 115, 211-252 (2015).
- 10 Goodfellow, I., Bengio, Y. & Courville, A. 6.2. 2.3 softmax units for multinoulli output distributions. Deep learning 180 (2016).

- 11 Brownlee, J. Probability for machine learning: Discover how to harness uncertainty with Python. (Machine Learning Mastery, 2019).
- 12 Loshchilov, I. & Hutter, F. Sgdr: Stochastic gradient descent with warm restarts. arXiv preprint arXiv:1608.03983 (2016).
- 13 Loshchilov, I. & Hutter, F. Decoupled weight decay regularization. arXiv preprint arXiv:1711.05101 (2017).
- 14 Loshchilov, I. & Hutter, F. Fixing weight decay regularization in adam. (2018).
- 15 Zhang, H., Cisse, M., Dauphin, Y. N. & Lopez-Paz, D. mixup: Beyond empirical risk minimization. arXiv preprint arXiv:1710.09412 (2017).
- 16 Lin, T.-Y., Goyal, P., Girshick, R., He, K. & Dollár, P. in Proceedings of the IEEE international conference on computer vision. 2980-2988.
- 17 Huang, G., Liu, Z., Van Der Maaten, L. & Weinberger, K. Q. in Proceedings of the IEEE conference on computer vision and pattern recognition. 4700-4708.

Again, we appreciate all of your insightful comments. We worked hard to be responsive to them. Thank you for taking the time and energy to help us improve the paper.

Sincerely yours,

Chang Chen

Department of Thoracic Surgery

Shanghai Pulmonary Hospital

Tongji University, School of Medicine

REVIEWERS' COMMENTS

Reviewer #1 (Remarks to the Author):

The authors submitted the revised manuscript, which was revised based on reviewers' comments. I checked every revised point as well as "Comments and Answer".

The authors revised or made additional comments/analysis. Most of these are reasonable and well considered. Some of them could not be solved at present, however I postulate this is due to the limitation of the present work and expected to be solved in next project.

Reviewer #2 (Remarks to the Author):

I thank the authors for thoroughly responding to my comments. With the edits made in the revised version, I am ready to recommend this manuscript for publication, given that a few more small issues are addressed:

- In Fig 3 & 4 legends and Statistical Analysis section, the authors refer to the multiple comparison correction method as "Benjaminiand Hochberg", whereas it should be Benjamini-Hochberg or "Benjamini and Hochberg"
- In "Variables associated with ONM" authors identified some variables as independent predictors based on their p-values, which involves comparison and therefore should be corrected
- References should be double-checked, e.g. [10] is missing from the main text references, while some supplementary references miss article titles; reference for PyTorch is missing.

RESPONSE TO REVIEWERS' COMMENTS

We would like to express our gratitude to each of the external reviewers for careful and thorough reading of this manuscript and for the thoughtful comments and constructive suggestions, which help to improve the quality of this study. The comments are encouraging and the reviewers appear to share our judgement that this study and its results are clinically important. Please see below, in **blue**, our detailed response to reviewers' comments (comments are in *italics*). All mentioned line numbers refer to the manuscript file with tracked changes. We hope the revised manuscript is acceptable for publication in *Nature Communications*

Reviewer 1:

1. *In Fig 3 & 4 legends and Statistical Analysis section, the authors refer to the multiple comparison correction method as "Benjaminiand Hochberg", whereas it should be Benjamini–Hochberg or "Benjamini and Hochberg".*

Reply:

We thank the reviewer for this important comment and we apologize for our terrible mistake. In this revision, we have corrected the description as "Benjamini and Hochberg" throughout the paper.

Changes in the text:

Method section Line 436-437: The Benjamini and Hochberg method was utilized to correct p values from multiple comparisons.

Figure legends section Line 634-635: p values from Delong's tests were adjusted by the

Benjamini and Hochberg corrections for 5 multiple comparisons.

Figure legends section Line 645-646: p values from Delong's tests were adjusted by the Benjamini and Hochberg corrections for 5 multiple comparisons.

Figure legends section Line 674: p values were adjusted by the Benjamini and Hochberg corrections.

2. In "Variables associated with ONM" authors identified some variables as independent predictors based on their p-values, which involves comparison and therefore should be corrected.

Reply:

Thank you for this significant suggestion. Independent predictors were determined based on the results of multivariable analyses, and multivariable analyses involves comparison but univariable analyses did not involve comparison. Therefore, in this revision, p values of multivariable analyses reported in the main tables and supplementary tables were all corrected by the Benjamini and Hochberg methods. After corrections, as displayed in Table 2, in the training set, a younger age (odds ratio [OR]: 0.967, 95% confidence interval [CI]: [0.951, 0.984], adjusted $p < 0.001$), pure solid type (OR: 2.525, 95% CI: [1.638, 3.891], adjusted $p < 0.001$), left location (OR: 1.512, 95% CI: [1.088, 2.100], adjusted $p = 0.023$), and central location (OR: 1.743, 95% CI: [1.202, 2.530], adjusted $p = 0.007$) were identified as independent predictors for occult N1 metastasis, and the pure solid type (OR: 3.389, 95% CI: [1.999, 5.745], adjusted $p < 0.001$) was independently related to occult N2 involvement.

Changes in the text:

Result section Line 128-134: As displayed in Table 2, in the training set, a younger age (odds ratio [OR]: 0.967, 95% confidence interval [CI]: [0.951, 0.984], adjusted p<0.001), pure solid type (OR: 2.525, 95% CI: [1.638, 3.891], adjusted p<0.001), left location (OR: 1.512, 95% CI: [1.088, 2.100], adjusted p=0.023), and central location (OR: 1.743, 95% CI: [1.202, 2.530], adjusted p=0.007) were identified as independent predictors for occult N1 metastasis, and the pure solid type (OR: 3.389, 95% CI: [1.999, 5.745], adjusted p<0.001) was independently related to occult N2 involvement.

Table 2. Logistic analyses of occult N1 and N2 metastasis before incorporation of the DLNMS for patients in the training set

Variables	Occult N1			Occult N2		
	Univariable		Multivariable	Univariable		Multivariable
	OR (95% CI)	p value	OR (95% CI)	OR (95% CI)	p value	adjusted p value
Age	0.979 (0.963-0.994)	0.008	0.967 (0.951-0.984)	0.992 (0.973-1.012)	0.445	
Sex (Male)	1.929 (1.404-2.652)	<0.001	1.403 (1.001-1.990)	1.463 (1.008-2.125)	0.046	0.985
Smoking history (Ever)	1.152 (0.824-1.470)	0.855		1.301 (0.878-1.353)	0.765	
Radiological type (Solid)	4.005 (2.718-5.903)	<0.001	2.525 (1.638-3.891)	4.231 (2.614-6.848)	<0.001	<0.001
Location (Left)	1.429 (1.048-1.949)	0.024	1.512 (1.088-2.100)	1.249 (0.862-1.810)	0.241	
Location (Central)	2.998 (2.146-4.188)	<0.001	1.743 (1.202-2.530)	1.936 (1.282-2.924)	0.002	0.430
Radiological size	1.385 (1.239-1.548)	<0.001	1.146 (1.001-1.313)	1.269 (1.124-1.433)	<0.001	0.100
SUVmax	1.127 (1.095-1.160)	<0.001	1.045 (0.805-1.356)	1.094 (1.060-1.129)	<0.001	0.473
MTV	1.001 (0.993-1.010)	0.746		0.999 (0.986-1.011)	0.845	
TLG	1.001 (1.000-1.002)	0.105		1.000 (1.000-1.001)	0.242	

DLNMS, deep learning nodal metastasis signature; SUV, standard uptake value; MTV, metabolic tumor volume; TLG, total lesion glycolysis; HR, hazard ratio; CI, confidence interval; p values of multivariable analyses were corrected by the Benjamini and Hochberg method.

Supplementary Table 1. Logistic analyses of occult N1 and N2 metastasis before incorporation of the DLNMS for patients in the validation set, external cohort and prospective cohort

Variables	Occult N1				Occult N2			
	Univariable		Multivariable		Univariable		Multivariable	
	OR (95% CI)	p value	OR (95% CI)	adjusted p value	OR (95% CI)	p value	OR (95% CI)	adjusted p value
Age	0.985 (0.970-1.001)	0.066	0.969 (0.952-0.986)	<0.001	1.010 (0.991-1.029)	0.318		
Sex (Male)	1.885 (1.396-2.544)	<0.001	2.110 (1.489-2.989)	<0.001	0.940 (0.671-1.317)	0.720		
Smoking history (Ever)	1.123 (0.733-1.305)	0.799			1.117 (0.821-1.203)	0.841		
Radiological type (Solid)	4.810 (3.240-7.141)	<0.001	6.886 (5.274-9.101)	<0.001	3.161 (2.092-4.776)	<0.001	8.697 (4.559-16.591)	<0.001
Location (Left)	1.149 (0.892-1.599)	0.234			1.126 (0.803-1.579)	0.490		
Location (Central)	2.962 (2.157-4.068)	<0.001	0.950 (0.647-1.395)	0.952	1.516 (1.019-2.255)	0.040	1.419 (1.172-1.718)	0.070
Tumor size	1.258 (1.150-1.376)	<0.001	1.084 (0.965-1.217)	0.261	1.159 (1.047-1.284)	0.004	1.097 (0.949-1.269)	0.281
SUVmax	1.114 (1.086-1.142)	<0.001	0.960 (0.700-1.317)	0.800	1.085 (1.055-1.115)	<0.001	0.800 (0.529-1.211)	0.291
MTV	0.997 (0.987-1.007)	0.569			0.989 (0.959-1.022)	0.552		
TLG	1.000 (1.000-1.001)	0.300			1.000 (0.999-1.001)	0.805		

DLNMS, deep learning nodal metastasis signature; SUV, standard uptake value; MTV, metabolic tumor volume; TLG, total lesion glycolysis; HR, hazard ratio; CI, confidence interval; p values of multivariable analyses were corrected by the Benjamini and Hochberg method.

Supplementary Table 2. Logistic analyses of occult N1 and N2 metastasis after incorporation of the DLNMS for patients in the training set

Variables	Occult N1				Occult N2			
	Univariable		Multivariable		Univariable		Multivariable	
	OR (95% CI)	p value	OR (95% CI)	adjusted p value	OR (95% CI)	p value	OR (95% CI)	adjusted p value
Age	0.979 (0.963-0.994)	0.008	0.972 (0.946-1.000)	0.147	0.992 (0.973-1.012)	0.445		
Sex (Male)	1.929 (1.404-2.652)	<0.001	0.939 (0.525-1.677)	0.935	1.463 (1.008-2.125)	0.046	0.936 (0.589-1.488)	0.936
Smoking history (Ever)	1.152 (0.824-1.470)	0.855			1.301 (0.878-1.353)	0.765		
Radiological type (Solid)	4.005 (2.718-5.903)	<0.001	0.597 (0.271-1.317)	0.362	4.231 (2.614-6.848)	<0.001	1.089 (0.549-2.161)	0.807
Location (Left)	1.429 (1.048-1.949)	0.024	1.017 (0.595-1.737)	0.952	1.249 (0.862-1.810)	0.241		
Location (Central)	2.998 (2.146-4.188)	<0.001	0.568 (0.254-1.063)	0.259	1.936 (1.282-2.924)	0.002	0.726 (0.440-1.197)	0.630
Tumor size	1.385 (1.239-1.548)	<0.001	1.370 (1.086-1.727)	0.036	1.269 (1.124-1.433)	<0.001	1.040 (0.862-1.254)	0.999
SUVmax	1.127 (1.095-1.160)	<0.001	0.912 (0.598-1.391)	0.861	1.094 (1.060-1.129)	<0.001	0.842 (0.474-1.368)	0.848
MTV	1.001 (0.993-1.010)	0.746			0.999 (0.986-1.011)	0.845		
TLG	1.001 (1.000-1.001)	0.065	1.001 (0.999-1.002)	0.429	1.000 (1.000-1.001)	0.242		
DLNMS	1307.850 (593.604-2881.506)	<0.001	2382.108 (885.311-6409.545)	<0.001	128.017 (64.132-255.540)	<0.001	206.552 (92.735-460.062)	<0.001

DLNMS, deep learning nodal metastasis signature; SUV, standard uptake value; MTV, metabolic tumor volume; TLG, total lesion glycolysis; HR, hazard ratio; CI, confidence interval; p values of multivariable analyses were corrected by the Benjamini and Hochberg method.

Supplementary Table 3. Logistic analyses of occult N1 and N2 metastasis after incorporation of the DLNMS for patients in the validation set, external cohort and prospective cohort

Variables	Occult N1				Occult N2			
	Univariable		Multivariable		Univariable		Multivariable	
	OR (95% CI)	p value	OR (95% CI)	adjusted p value	OR (95% CI)	p value	OR (95% CI)	adjusted p value
Age	0.985 (0.970-1.001)	0.066	0.975 (0.952-0.999)	0.157	1.010 (0.991-1.029)	0.318		
Sex (Male)	1.885 (1.396-2.544)	<0.001	1.586 (0.968-2.601)	0.156	0.940 (0.671-1.317)	0.720		
Smoking history (Ever)	1.123 (0.733-1.305)	0.799			1.117 (0.821-1.203)	0.841		
Radiological type (Solid)	4.810 (3.240-7.141)	<0.001	1.066 (0.535-2.152)	0.842	3.161 (2.092-4.776)	<0.001	1.949 (0.655-2.774)	0.648
Location (Left)	1.149 (0.892-1.599)	0.234			1.126 (0.803-1.579)	0.490		
Location (Central)	2.962 (2.157-4.068)	<0.001	0.646 (0.208-1.452)	0.357	1.516 (1.019-2.255)	0.040	0.572 (0.214-1.147)	0.288
Tumor size	1.258 (1.150-1.376)	<0.001	1.155 (0.980-1.360)	0.149	1.159 (1.047-1.284)	0.004	1.004 (0.832-1.210)	0.971
SUVmax	1.114 (1.086-1.142)	<0.001	0.971 (0.642-1.468)	0.888	1.085 (1.055-1.115)	<0.001	0.703 (0.428-1.156)	0.275
MTV	0.997 (0.987-1.007)	0.569			0.989 (0.959-1.022)	0.552		
TLG	1.000 (1.000-1.001)	0.300			1.000 (0.999-1.001)	0.805		
DLNMS	508.761 (274.343-943.480)	<0.001	435.448 (216.951-873.998)	<0.001	69.339 (39.648-121.266)	<0.001	66.550 (36.697-120.689)	<0.001

DLNMS, deep learning nodal metastasis signature; SUV, standard uptake value; MTV, metabolic tumor volume; TLG, total lesion glycolysis; HR,

hazard ratio; CI, confidence interval; p values of multivariable analyses were corrected by the Benjamini and Hochberg method.

3. *References should be double-checked, e.g. [10] is missing from the main text references, while some supplementary references miss article titles; reference for PyTorch is missing.*

Reply:

We appreciate the reviewer for these significant comments and we apologize for our terrible mistakes. In this revision, references in the main text and supplementary material were corrected and reference for PyTorch was added.

Changes in the text:

References section Line 477-479:

10 National Comprehensive Cancer Network. (NCCN) Clinical Practice Guidelines in Oncology. Non-Small Cell Lung Cancer, Version 1. 2022. Available at: https://www.nccn.org/professionals/physician_gls/default.aspx. Accessed 7 Dec 2021.

Supplementary Method 13. References:

- 1 Hariharan, B., Arbeláez, P., Girshick, R. & Malik, J. Hypercolumns for object segmentation and fine-grained localization. Proceedings of the IEEE conference on computer vision and pattern recognition, 447-456 (2015).
- 2 Lee, K., Zung, J., Li, P., Jain, V. & Seung, H. S. Superhuman accuracy on the SNEMI3D connectomics challenge. arXiv preprint arXiv:1706.00120 (2017).
- 3 Quan, T., Hildebrand, D. & Jeong, W. Fusionnet: A deep fully residual convolutional neural network for image segmentation in connectomics. arXiv 2016. arXiv preprint arXiv:1612.05360 (2016).
- 4 Beier, T. et al. Multicut brings automated neurite segmentation closer to human performance. Nature methods 14, 101-102 (2017).

- 5 Toubal, I. E., Duan, Y. & Yang, D. Deep learning semantic segmentation for high-resolution medical volumes. 2020 IEEE Applied Imagery Pattern Recognition Workshop (AIPR), 1-9 (2020).
- 6 Zeng, T., Wu, B. & Ji, S. DeepEM3D: approaching human-level performance on 3D anisotropic EM image segmentation. *Bioinformatics (Oxford, England)* 33, 2555-2562 (2017).
- 7 Simonyan, K. & Zisserman, A. Very deep convolutional networks for large-scale image recognition. *arXiv preprint arXiv:1409.1556* (2014).
- 8 He, K., Zhang, X., Ren, S. & Sun, J. Deep residual learning for image recognition. *Proceedings of the IEEE conference on computer vision and pattern recognition*, 770-778 (2016).
- 9 Russakovsky, O. et al. Imagenet large scale visual recognition challenge. *International journal of computer vision* 115, 211-252 (2015).
- 10 Goodfellow, I., Bengio, Y. & Courville, A. 6.2. 2.3 softmax units for multinoulli output distributions. *Deep learning* 180 (2016).
- 11 Brownlee, J. *Probability for machine learning: Discover how to harness uncertainty with Python.* (Machine Learning Mastery, 2019).
- 12 Loshchilov, I. & Hutter, F. Sgdr: Stochastic gradient descent with warm restarts. *arXiv preprint arXiv:1608.03983* (2016).
- 13 Loshchilov, I. & Hutter, F. Decoupled weight decay regularization. *arXiv preprint arXiv:1711.05101* (2017).
- 14 Loshchilov, I. & Hutter, F. Fixing weight decay regularization in adam. *arXiv preprint arXiv:1711.05101* (2018).
- 15 Zhang, H., Cisse, M., Dauphin, Y. N. & Lopez-Paz, D. mixup: Beyond empirical risk minimization. *arXiv preprint arXiv:1710.09412* (2017).

- 16 Lin, T.-Y., Goyal, P., Girshick, R., He, K. & Dollár, P. Focal loss for dense object detection. Proceedings of the IEEE international conference on computer vision, 2980-2988 (2017).
- 17 Paszke, A. et al. Pytorch: An imperative style, high-performance deep learning library. Advances in neural information processing systems 32 (2019).
- 18 Huang, G., Liu, Z., Van Der Maaten, L. & Weinberger, K. Q. Densely connected convolutional networks. Proceedings of the IEEE conference on computer vision and pattern recognition, 4700-4708 (2017).

Again, we appreciate all of your insightful comments. We worked hard to be responsive to them. Thank you for taking the time and energy to help us improve the paper.

Sincerely yours,

Chang Chen

Department of Thoracic Surgery

Shanghai Pulmonary Hospital

Tongji University, School of Medicine